# Radar and ground-level measurements of precipitation collected by EPFL during the ICE-POP 2018 campaign in South-Korea

Josué Gehring[1], Alfonso Ferrone[1], Anne-Claire Billault–Roux[1], Nikola Besic[2], Kwang Deuk Ahn[3], GyuWon Lee[4], and Alexis Berne[1]

[1]Environmental Remote Sensing Laboratory, École Polytechnique Fédérale de Lausanne (EPFL), Lausanne, Switzerland
[2]Centre Météorologie Radar, Météo France, Toulouse, France
[3]Department of Astronomy and Atmospheric Sciences, Kyungpook National University, Daegu, Korea
[4]Numerical data application division, Numerical modeling center, Korea Meteorological Administration, Seoul, Korea

**Correspondence:** Alexis Berne (alexis.berne@epfl.ch)

**Abstract.** This article describes a four-month dataset of precipitation and cloud measurements collected during the International Collaborative Experiments for PyeongChang 2018 Olympic and Paralympic winter games (ICE-POP 2018). This paper aims to describe the data collected by the Environmental Remote Sensing Laboratory of the École Polytechnique Fédérale de Lausanne. The dataset includes observations from an X-band dual-polarisation Doppler radar, a W-band Doppler cloud profiler, a multi-angle snowflake camera and a two-dimensional video disdrometer (https://doi.pangaea.de/10.1594/PANGAEA.918315, Gehring et al. (2020a)). Classifications of hydrometeor types derived from dual-polarisation measurements and snowflake photographs are presented. The dataset covers the period from 15 November 2017 to 18 March 2018 and features nine precipitation events with a total accumulation of 195 mm of equivalent liquid precipitation. This represents 85% of the climatological accumulation over this period. To illustrate the available data, measurements corresponding to the four precipitation events with the largest accumulation are presented. The synoptic situations of these events were contrasted and influenced the precipitation type and accumulation. The hydrometeor classifications reveal that aggregate snowflakes were dominant and that some events featured significant riming. The combination of dual-polarisation variables and high-resolution Doppler spectra with ground-level snowflake images makes this dataset particularly suited to study snowfall microphysics in a region where such measurements were not available before.

## 1 Introduction

Precipitation measurements in mountainous regions are paramount to characterise the spatial distribution of precipitation and understand the effect of orography on microphysics. Korea's geographical environment provides a unique setting for precipitation studies: its location on a mountainous peninsula in the mid-latitudes is prone to large moisture advection by baroclinic systems and orographic lifting driving cloud and precipitation formation. Unlike other mountain ranges such as the Alps, the Rockies, the Olympic, the Cascade and the Coast Mountains (Bougeault et al., 2001; Saleeby et al., 2009; Houze et al., 2017; Stoelinga et al., 2003; Joe et al., 2010), the study of precipitation in the Taebaek Mountains in Korea has not been as extensive. Kim et al. (2018) investigated the microphysics of two snowfall events in the Taebaek mountains during the Experiment on

Snow Storms At Yeongdong (ESSAY) campaign using radiosoundings, snowflake images and numerical simulations. They suggested that future field campaigns should include dual-polarisation radars and a multi-angle snowflake camera (MASC) to better understand the microphysics of precipitation in this region. There was hence a need for a precipitation measurement campaign in the Taebaek mountains with remote sensing and in situ measurements.

5 Several past field campaigns demonstrated the usefulness of combined remote sensing and in situ measurements for snowfall studies. The TOSCA project in the Bavarian Alps in Germany (Löhnert et al., 2011) combined vertically pointing radars, radiometers and optical disdrometers among others, to better characterise the vertical distribution of snowfall for satellite retrievals and numerical model validations. During the 2015/16 fall–winter season, the OLYMPEX campaign (Houze et al., 2017) took place in the vicinity of the mountainous Olympic Peninsula, USA, to study how Pacific precipitation systems 10 are influenced by the orography. The BAECC field campaign (Petäjä et al., 2016) provided eight months of measurements in Hyytiälä, Finland, to study biogenic aerosols, clouds, and precipitation and their interactions. In-situ and remote sensing instruments have also proven very useful to study atmospheric radiation, cloud and precipitation properties in polar regions. The North Slope of Alaska atmospheric observatory (Verlinde et al., 2016) in Barrow and Oliktok provides a long series of measurements including radiometers, lidars, cloud radars and a MASC among many other instruments. Another major Arctic 15 measurement site is located at Summit, Greenland, where the ICECAPS (Shupe et al., 2013) field campaign was conducted to collect measurements of radiation, clouds and precipitation to study the energy and hydrological budgets of the Greenland Ice Sheet. In Antarctica, the APRES3 field campaign (Genthon et al., 2018) provided the first dual-polarisation radar measurements from November 2015 to February 2016. Along with snowflake photographs, micro rain radar and lidar measurements, the dataset led to unprecedented insights in Antarctic snowfall microphysics (Grazioli et al., 2017a). Finally, the AWARE campaign 20 (Lubin et al., 2020) gathered cloud radars, lidars, radiometers, aerosols and microphysical measurements from December 2015 to December 2016 in McMurdo Station, Antarctica. This unique dataset offers numerous cases for mixed-phase cloud parametrisation in weather and climate models. The measurements of these field campaigns allowed for innovative studies and new insights in cloud and precipitation processes in these various regions (Kalesse et al., 2016; von Lerber et al., 2017; Grazioli et al., 2017b; Cole et al., 2017; Zagrodnik et al., 2019). For a better understanding of cloud and precipitation processes 25 in the Taebaek mountains, a field campaign combining remote sensing and in situ measurements is needed. The PyeongChang 2018 Olympic and Paralympic winter games were the opportunity to initiate interest and collaboration for such a campaign. Indeed, accurate weather forecasts during Winter Olympic Games are an organisational need and a real scientific challenge. It is also a great opportunity to foster international collaboration and gather the atmospheric science community. One successful example of such a joint effort was the Science of Nowcasting Olympic Weather for Vancouver 2010 campaign, which led to 30 novel findings on precipitation (Thériault et al., 2012; Schuur et al., 2014; Berg et al., 2017) and nowcasting (Haiden et al., 2014), as well as to new instrumental developments (Boudala et al., 2014). Along the same line, the Korea Meteorological Administration organised the International Collaborative Experiments for Pyeongchang 2018 Olympic and Paralympic winter games (ICE-POP 2018). The main goals of ICE-POP 2018 were to support forecasters with high-resolution model simulations and radar data, as well as to gain more insight into orographic precipitation in the Taebaek mountains. For this purpose, remote

sensing and in-situ measurements of cloud and precipitation were conducted in the Gangwon-do province between November 2017 and May 2018.

This article presents the data collected by the Environmental Remote Sensing Laboratory of the École Polytechnique Fédérale de Lausanne during ICE-POP 2018. It includes measurements from an X-band dual-polarisation Doppler radar, a W-band Doppler cloud profiler, a multi-angle snowflake camera and a two-dimensional video disdrometer. Such a dataset is unique, as it includes multi-frequency radar and ground-based measurements in a region where similar measurements were scarce before ICE-POP 2018. As shown in Gehring et al. (2020b) it is a useful dataset for snowfall microphysics studies and could also be relevant for validation of numerical weather prediction models. Section 2 presents the campaign and the instrumental setup and section 3 describes the data processing. Section 4 illustrates the dataset with measurements and hydrometeor classifications corresponding to the four events with the largest accumulation. Section 5 closes this paper with concluding remarks.

## 2 Measurement sites and instruments

In this study, we will focus on the data collected by an X-band dual-polarisation Doppler (polarimetric) radar (MXPol), a W-band Doppler cloud profiler (WProf), a multi-angle snowflake camera (MASC) and a two-dimensional video disdrometer (2DVD). Figure 1 shows the location of the instruments. The first measurement site was located in the Gangneung Wonju national university (GWU) at 66 m a.s.l. The second measurement site was in Bokwang 1-ri community centre (BKC) 6 km inland from Gangneung at 175 m a.s.l. The third measurement site Mayhills (MHS) was located in the PyeongChang province at 789 m a.s.l. 19 km inland of GWU. Radiosoundings were launched by the Korea Meteorological Administration (KMA) in Daegwallyeong (DGW), 2 km from MHS.

## 2.1 X-band polarimetric radar: MXPol

The scanning X-band polarimetric radar, named MXPol, was installed in GWU on the coast of the East Sea. The main variables retrieved from MXPol measurements in dual-pulse pair (DPP) mode are the equivalent reflectivity factor at horizontal polarisation $Z_H$ (dBZ), the differential reflectivity $Z_{DR}$ (dB), the specific differential phase shift on propagation $K_{dp}$ ($°$km$^{-1}$), the copolar correlation coefficient $\rho_{hv}$, the mean Doppler velocity $V_D$ (m s$^{-1}$) and the Doppler spectral width $\sigma_v$ (m s$^{-1}$). Additionally, in fast-Fourier transform (FFT) mode, the full Doppler spectrum at 0.17 m s$^{-1}$ resolution is retrieved at each range gate. MXPol operates at 9.41 GHz with a typical angular resolution of 1$°$, range resolution of 75 m, non-ambiguous range of 120 km and a Nyquist velocity of 39 m s$^{-1}$ in DPP or 11 m s$^{-1}$ in FFT mode. Only the data up to 28 km range are saved, since the decrease in sensitivity and increase in sampling volume makes the further gates less relevant for microphysical studies. A more technical description of MXPol can be found in Schneebeli et al. (2013). The main scan cycle was composed of two hemispherical range height indicators (RHIs) at 227$°$ and 317$°$ azimuth in FFT and DPP modes, respectively. The former is towards MHS, while the latter is perpendicular to this direction following the coast as shown by the red dotted line in Fig. 1. The two RHIs were followed by one plan position indicator (PPI) in DPP mode at 6$°$ elevation. The cycle was either completed

by two other RHIs or PPIs depending on the event. The scan cycle had a 5 min duration and was repeated indefinitely. At least once an hour, a PPI at 90° elevation in FFT mode was performed for differential reflectivity calibration.

## 2.2 W-band cloud profiler: WProf

A W-band Doppler cloud profiler (WProf) was deployed at Mayhills site (MHS). WProf is a frequency modulated continuous wave (FMCW) radar operating at 94 GHz at a single polarisation. It allows to measure with different range and Doppler resolutions using typically three vertical chirps (Table 1). The main variables retrieved are the equivalent reflectivity factor $Z$, the mean Doppler velocity $V_D$, the Doppler spectral width $\sigma_v$ $(\mathrm{m\,s^{-1}})$, skewness and kurtosis. The full Doppler spectrum is also available. More details on WProf can be found in Küchler et al. (2017). WProf contains a 89 GHz radiometer, which can be used to retrieve the liquid water path (LWP) and the integrated water vapour (IWV). We computed LWP and IWV using the method described in Billault-Roux and Berne (2020). The brightness temperature measurements had a bias of 20 K, which we corrected. After correction the root mean squared error (RMSE) is 2.88 K. The RMSE of LWP and IWV are 86.5 $\mathrm{g\,m^2}$ and 1.72 $\mathrm{kg\,m^2}$ respectively taking radiosoundings as the reference. More information on the uncertainty of this algorithm can be found in Billault-Roux and Berne (2020). In particular, note that the accuracy is deteriorated in case of intense precipitation. WProf was calibrated by the manufacturer Radiometer Physics GmbH just before the ICE-POP 2018 campaign. This included a calibration of the 89 GHz radiometer with liquid nitrogen and a calibration of the radar with disdrometers following the method of Myagkov et al. (2020). The uncertainty of WProf reflectivity calibration is +/- 1 dB. Note that compared to Gehring et al. (2020b) the calibration was updated following RPG's recommendations and hence the $Z_e$ values of WProf do not match with those of this article. This calibration correction was applied on the data available on PANGAEA (Gehring et al., 2020a). The radomes were in good shape and were not changed between the calibration and the field campaign. The blowers, which prevent liquid water to accumulate on the radomes, were switched on all the time. The radar pointing was evaluated by checking the levels at the beginning and the end of the campaign. The levels shown that the radar was pointing almost perfectly vertically. However, since the vertical alignment was not monitored constantly, the spectral and Doppler velocity data should be interpreted carefully, especially in case of strong horizontal winds.

## 2.3 Multi-angle snowflake camera: MASC

A multi-angle snowflake camera (MASC) was deployed in a double fence windshield in MHS. The MASC is composed of three coplanar cameras separated by an angle of 36°. As hydrometeors fall in the triggering area, high-resolution stereographic pictures are taken and their fall velocities are measured. A complete description of the MASC can be found in Garrett et al. (2012). The MASC images were used as input parameters to a solid hydrometeor classification algorithm. Individual particles are classified into six solid hydrometeor types, namely small particles (SP), columnar crystals (CC), planar crystals (PC), a combination of column and plate crystals (CPC), aggregates (AG) and graupel (GR). In addition a degree of riming ranging from 0 to 1 is computed. A detailed explanation of the algorithm is provided in Praz et al. (2017). Note that the degree of riming of melting particles and raindrops computed by this method is usually quite high. One should not consider the degree of riming for particles with a melting probability higher than 50 %.

## 2.4 Two-dimensional video disdrometer: 2DVD

A two-dimensional video disdrometer (2DVD) was deployed in BKC. A detailed description of the instrument can be found in Kruger and Krajewski (2002) and Schönhuber et al. (2007). Here we will describe the general measurement principle. Two orthogonal light sources are projected onto a line-scan camera. Particles falling through the light sheets project a one-dimensional section on the photodetectors. Theses one-dimensional profiles are then combined to form a two-dimensional view of the particle. Horizontal wind induces a horizontal displacement of the particles, such that the superposition of the one-dimensional sections can lead to distorted particles. This issue is thoroughly investigated with numerical simulations in Nešpor et al. (2000). The two orthogonal two-dimensional projections yield to a three-dimensional shape information, which can be used to compute the equivalent drop diameter and the aspect ratio. This makes it possible to compute the raindrop size distribution (DSD). Since the vertical distance between the two light sheets is known, the particles' fall velocities can also be computed. 2DVD data can also be used for snowfall microphysics studies. Brandes et al. (2007) derived the particle size distribution (PSD) from 2DVD data in Colorado. Huang et al. (2010) and Huang et al. (2015) used a 2DVD to derive radar reflectivity–snowfall rate relations. Finally Grazioli et al. (2014) used 2DVD data to develop a supervised hydrometeor classification method.

## 3 Data processing

### 3.1 MXPol

First, the noise floor is determined from the raw power following the method from Hildebrand and Sekhon (1974). Then, the polarimetric variables are computed based on the backscattering covariance matrix following Doviak and Zrnic (1993). The computation of $K_{dp}$ is based on an ensemble of Kalman filters as detailed in Schneebeli et al. (2014).

#### 3.1.1 Calibration

To monitor the stability of the radar signal, a radar target simulator (RTS, http://www.palindrome-rs.ch/products/radar-target-simulator/, last access: September 4th, 2020) developed by Palindrome Remote Sensing GmbH was installed during the campaign. Unfortunately, due to technical issues during the campaign, the data could not be used for calibration of the radar. However, we conducted dedicated calibration measurements with the RTS in July 2018 just after the ICE-POP 2018 campaign. The results showed that the reflectivity measurements have errors smaller than 1 dBZ.

#### 3.1.2 Hydrometeor classification

The dual-polarisation observables were used to feed the hydrometeor classification from Besic et al. (2016). The centroids of all four polarimetric variables used for the classification have been trained on MXPol data from various field campaigns in the Swiss Alps, in Ardèche (France), in Antarctica and on the present dataset in Korea. Recently, Besic et al. (2018) developed a de-mixing approach of this hydrometeor classification, in which the proportion of hydrometeors for each radar sampling volume

is estimated, instead of one dominant class. The classes are crystals, aggregates, light rain, rain, rimed ice particles, vertically aligned ice, wet snow, ice hail and high-density graupel and melting hail. This approach is essentially built upon the concept of entropy, as defined in Besic et al. (2016), which reflects the uncertainty with which a hydrometeor class is assigned to one sampling volume. This de-mixing method has the advantage of revealing the spectrum of hydrometeors present in the observed

precipitation. The classification was applied to all RHIs of the precipitation events shown in Fig. 4. Only the data above 2000 m a.s.l. have been selected for the hydrometeor classification shown in Sect. 4, because of ground echoes contamination and partial beam filling below this altitude.

### 3.1.3  Differential reflectivity bias correction

For a correct interpretation of $Z_{DR}$, the offset introduced by the existence of differences in amplitude in the horizontal and

vertical channels needs to be subtracted. This calibration can be achieved by analysing $Z_{DR}$ values in a specific subset of the range gates of the vertical PPI, which were performed at least once per hour during the whole campaign. Unfortunately, MXPol is affected by extremely high $Z_{DR}$ values in the low gates, probably caused by issues on the transmit-receive limiter. Therefore, a classical calibration procedure such as the one described in Gorgucci et al. (1999) cannot be applied. Instead, we decided to select the range interval used for the correction among the upper gates, unaffected by the issue. The first step of the

calibration procedure was the removal of data with signal to noise ratios lower than 5 dB or $\rho_{hv} < 0.95$. For each PPI, we also removed the range gates in which we encountered at least one non-valid $Z_{DR}$ measurement, to avoid introducing a bias caused by some angles being over-represented. Subsequently, we computed, for each range gate, the standard deviation of the $Z_{DR}$ distribution over the whole campaign duration. This standard deviation is remarkably constant more than 1 km above the radar, while its magnitude increases rapidly in the closest gates, due to the issue mentioned before. After computing the median of

these values in the top 25 % of the range gates, we impose a maximum threshold of 0.1 dB on the absolute difference between the standard deviation at each range gate and the median value. The median of all $Z_{DR}$ values from the range gates satisfying the condition is 2.66 dB with 50 % of the values within 0.32 dB. This median value of 2.66 dB was subtracted from all $Z_{DR}$ measurements to get the corrected $Z_{DR}$ dataset.

### 3.2  MASC

The raw data from the MASC are stereographic photographs of hydrometeors and measurements of fall velocities. Praz et al. (2017) developed a hydrometeor classification and riming degree estimation of MASC pictures based on a multinomial logistic regression model. More recently, Hicks and Notaroš (2019) used convolutional neural networks to classify MASC snowflake images. In this paper, we will use the algorithm from Praz et al. (2017) to classify the MASC data collected during ICE-POP 2018. We will show the hydrometeor classification, as well as the riming degree and melting probability results. Note that the

riming degree of small particles is not reliable, since it is computed over a few pixels only. Therefore we discard small particle in the time series of riming degree (i.e. Figs 6,8,10,12). In addition, raindrops appear as small bright spots in MASC images (reflection of flashes) and are hence classified as small particles. Therefore, removing small particles from the riming degree statistics avoid the bias related to raindrops.

Schaer et al. (2020) developed a method to classify MASC images as blowing snow, precipitation or a mixture of those. This makes it possible to filter the results and minimise the influence of possible blowing snow. Even though a double fence wind-shield was present during the ICE-POP 2018 campaign, 31% of the particles were classified either as a mixture of precipitation and blowing snow or pure blowing snow. In the present dataset, all particles are retained, but the information needed to filter out blowing snow particles is added. As explained in Schaer et al. (2020) a threshold of 0.193 on the normalised angle $\psi$ can be used with $\psi < 0.193$ corresponding to pure precipitation. The results of the hydrometeor classification shown in Section 4 correspond to pure precipitation only.

## 3.3 WProf

The raw data from WProf were saved without any filtering, in the form of raw Doppler spectra. The spectra are then dealiased with an algorithm based on the minimisation of the spectral width at each range gate, similar to Ray and Ziegler (1977). This method assumes aliasing up to one folding, which is sufficient for the Nyquist intervals considered here (Table 1). From the dealiased spectra, the noise floor was determined using the method from Hildebrand and Sekhon (1974). The moments ($V_D$, $Z$, $\sigma_v$, skewness and kurtosis) are then computed from the dealiased spectra above the noise floor.

### 3.3.1 Atmospheric gas attenuation

To correct for attenuation due to atmospheric gases, we used the Passive and Active MicrowaveTRansfer Model (PAMTRA Mech et al., 2020) available at https://github.com/igmk/pamtra (last access: May 27th, 2020) and humditiy, temperature and pressure profiles from radiosoundings launched at DGW. Radiosoundings were usually available every three hours, but sometimes up to twelve hours. In order to quantify the temporal variability, we computed the variogram of IWV from all radiosoundings and concluded that the decorrelation time is long enough so we can expect relatively accurate interpolated values in between radiosoundings. We hence decided to compute a linear interpolation between the two nearest radiosoundings in time to get the profiles at a 5 min resolution, which was then used to compute the gas attenuation and applied to each WProf profile. Figure 2 shows the histograms of dry air, water vapour and total two-way attenuation at W-band from 30 November 2017 to 31 March 2018, which corresponds to the period during which radiosoundings are available. Outside of this period, the WProf reflectivity measurements were not corrected for attenuation. For dry air the values range from about 0.26 dB to 0.33 dB. For water vapour, the values range from nearly 0 dB to 1.75 dB. The attenuation depends on the absolute humidity and hence the range of values is larger, going from a very dry air to a saturated environment. The total two-way attenuation (up to 10 km) varies between 0.3 dB to 2 dB.

## 3.4 Sensitivity

To visualise the sensitivity of WProf and MXPol, Fig. 3 shows the empirical joint distributions of range and reflectivity values during all precipitation events of the ICE-POP 2018 campaign. The minimum measured reflectivity values represent the sensitivity. A threshold on the signal to noise ratio of 0 dB was applied on MXPol and WProf data in all figures presented.

For WProf (Fig. 3a) we can clearly see the effect of the three vertical chirps on the minimum detectable reflectivity. One can see that WProf has a higher sensitivity than MXPol at all range gates.

## 4    Precipitation events

Figure 4 shows precipitation and temperature information during the precipitation events. Table 3 shows the exact date and time of the different events. The atmospheric conditions during the ICE-POP 2018 campaign were climatologically cold and dry. The winter 2018 (December 2017 – February 2018) had a total precipitation accumulation of 93 mm in MHS, while the climatological value (KMA, 2011) is 153 mm in Daegwallyeong, 2 km from MHS. The major precipitation event on 28 February 2018 contributed to 62% of the winter 2018 precipitation accumulation, which shows that the rest of the winter was extremely dry (36 mm excluding the 28 February event). March featured a few significant precipitation events leading to 83 mm of precipitation accumulation, while the climatological value is 76 mm. We have four main events, which we will present in this section: 25 November 2017, 28 February 2018, 04 and 07 March 2018. These events stand out out by their significant precipitation accumulation. Table 4 shows the amount of data collected by each instrument. The measurement time from MXPol does not take into account the repositioning of the antenna between each scan, which typically takes the same time as the scan averaged over the whole cycle. This is why the measurement duration from MXPol is about half that from WProf, which measured continuously. The number of triplets captured by the MASC indicates the number of sets of three pictures captured by the three cameras. For each picture, the classification selects one particle that is in focus. The maximum rate of images is 2 Hz, hence only two hydrometeors can be identified every second. The 2DVD measures continuously at a rate of 34.1 kHz and can identify multiple particles in its sampling area, unlike the MASC. This explains why the number of particles captured by the 2DVD is two order of magnitudes greater than the number of triplets of the MASC.

### 4.1    25 November 2017

The 25 November 2017 event has the third-largest precipitation accumulation, but the second-largest mean precipitation rate (see Fig. 4). Figure 5 shows a strong westerly flow associated with a broad upper-level trough. Analysis of backward trajectories (not shown) revealed that the moisture was pumped from the Yellow Sea and lifted over the topography leading to a broad cloud and precipitation system.

Figure 6a shows the reflectivity measured by WProf. The precipitating cloud is shallow with a cloud top at 4800 m. Precipitation started at 08:00 UTC and lasts until 16:00 UTC. Figure 6b shows the Doppler velocity measured by WProf. Negative (positive) values represent a relative displacement towards (away from) the radar. Except for some local turbulence below 2000 m, there is no significant updrafts in the cloud. Note that the precipitation rate data shown in Fig. 6c are not part of the presented dataset, but can be requested from the contact author. One can observe that rimed particles dominate below 3000 m a.s.l. from 11:30 to 13:30 UTC (dominance of red shades in Fig. 6d).

Figure 6e shows a time series of the classification from the MASC. At this time, the MASC was in BKC at only 175 m a.s.l. and it observed almost exclusively raindrops, which are classified as small particles (Praz et al., 2017). Figure 6f shows

the DSD computed from 2DVD data at BKC. The largest raindrops are observed during the most intense precipitation period (11:00–13:00 UTC) and correspond to the highest vertical extension of the cloud.

## 4.2   28 February 2018

The 28 February 2018 event stands out as the most intense of the whole campaign, in terms of accumulation and mean precipitation rate. At 00:00 UTC (not shown) a prominent PV streamer on eastern China and a low-pressure system eastward over the Yellow Sea are present. The PV streamer intensifies the surface cyclone and by 12:00 UTC 28 February (Fig. 7) the system is fully developed with the warm front passing over PyeongChang and leading to the observed precipitation. Note that the cyclone intensified by 25 hPa between 27 February, 18:00 UTC and 28 February, 12:00 UTC due to the upper-level forcing from the PV streamer. This event is presented in more details in Gehring et al. (2020b).

At 00:00 UTC 28 February, the nimbostratus (i.e. precipitating cloud associated with the warm front) can be observed above 2000 m, while fog is forming below 1000 m (Fig. 8a). Between the fog and the nimbostratus base, a dry layer is present where the precipitation from the nimbostratus sublimates to form virgas. At 03:00 UTC precipitation reaches the ground and lasts until 16:00 UTC. As temperatures at MHS are between 0 and 2 ° C before 06:00 UTC, the liquid water attenuation of the melting snowflakes can lead to underestimation of the reflectivity measurements of both MXPol and WProf. After 06:00 UTC, the temperatures at MHS are below freezing (Fig.8c) and hence there is almost no liquid water attenuation (attenuation from SLW droplets can be neglected). One can notice a region of embedded convection between 07:30 and 08:00 UTC and turbulence around 4000 m between 08:00 and 10:00 UTC (Fig.8b). The two first maxima of precipitation rate around 06:00 and 10:00 UTC (Fig.8c) correlates well with regions of high reflectivity extending up to about 4000 m a.s.l. During the passage of the front aggregates prevail, while from 06:00 to 08:00 UTC rimed particles dominate (Fig.8d).

Figure 8e shows the time series of the MASC classification. There are periods of missing data because the cameras were covered with rime. The event was dominated by aggregates, except at the end, where the temperature was colder and graupel and small particles are present. Note that the classification of Praz et al. (2017) classifies only fully rimed particles as graupel. The class aggregates also contain rimed particles, which explained why the period dominated by rimed particles in Fig. 8d (06:00 to 08:00 UTC) is not visible in the MASC classification as graupel particles. However, it is clear from the degree of riming, that rimed particles are present during this period.

## 4.3   04 March 2018

The 04 March 2018 event has the second-largest precipitation accumulation. Figure 9 shows the synoptic conditions. There is a strong south-westerly flow advecting significant moisture from the Yellow Sea, as can be seen by the integrated vapour flux (brown arrows) fluxes reaching 1000 $\mathrm{kg\,m^{-1}\,s^{-1}}$ and a low-pressure system located south of Korea. This large moisture transport leads to widespread precipitation over the Korean peninsula with a maximum over the centre of South Korea. The equivalent potential temperature shows the presence of warm and humid air reaching the cyclone's centre. The large sea-level pressure gradient on the eastern Korean coast suggests the presence of strong easterly winds. This easterly flow impinging

the Taebaek mountains from the East Sea might have been orographically lifted and participated in an enhancement of the observed precipitation.

Figure 10a,b shows the reflectivity and Doppler velocity from WProf. The beginning of the event is dominated by rain with a melting layer around 2500 m which appears clearly from the Doppler velocity (i.e. the sharp gradient in Doppler velocity showing the transition from snowflakes to faster falling raindrops). One can see the attenuation in the rain as a sharp decrease in $Z_e$ above 2000 m around 12:00 to 14:00 UTC. The melting layer abruptly drops to the ground level (i.e 789 m a.s.l. at MHS) at 14:00 UTC as temperatures quickly dropped below 0 °C (Fig. 10c). The cloud contains mainly crystals and aggregates (Fig. 10d), but also some rimed particles above the rain between 12:00 and 14:00 UTC.

Figure 10e shows the time series from the MASC classification. One can see that small particles (i.e. raindrops in this case) are dominating until just before 14:00 UTC. Aggregates are then the dominant hydrometeor type, apart from small particles. Graupel particles are more numerous compared to the previous events. The data gap in the 2DVD from about 20:30 UTC to 00:00 UTC, 04 March 2018 is due to a technical issue with the instrument.

## 4.4   07 March 2018

The 07 March 2018 event was the fourth most important in terms of precipitation accumulation (see Figure 4), but was the longest one because of a shallow precipitating system which lasted 12 hours after the main part of the event. On 07 March 00:00 UTC an upper-level trough is moving eastwards from China. Korea is under the influence of a ridge and clear sky conditions dominate. As the trough moves, moist unstable air from the Yellow Sea is advected over Korea and precipitation sets in. Starting from 07 March 15:00 UTC a low-pressure system develops south of the Korean peninsula and the trough becomes a broad PV streamer. The precipitation intensity increases until the PV streamer passes over Korea. At 18:00 UTC the precipitation weakens, while the low-pressure system is further intensifying on the eastern flank of the PV streamer and reaches Japan with more intense precipitation than observed in Korea. The key differences between this event and the 28 February are that the cyclone formed more to the east with a less pronounced PV streamer and that the locations of both features were not appropriate for a mutual intensification, as was the case on 28 February. This suggests that the timing and respective positions of the PV streamer and the low-pressure system during the 28 February event were key ingredients for its intensity.

Figure 12a,b shows the reflectivity and Doppler velocity from WProf. The nimbostratus cloud associated with the surface cyclone generates precipitation, which starts around 10:00 UTC and lasts until 04:00 UTC. A shallower precipitating system brings again precipitation from around 08:30 UTC to 19:00 UTC. This event has some similarities with the 28 February: they are both associated with a surface cyclone at the eastern flank of a PV streamer and they both feature a nimbostratus cloud followed by a shallower precipitating system. The latter is also associated with graupel particles (Fig.12e). The radar-based classification (Fig. 12d) shows mainly crystals and some aggregates. Since only values above 2000 m are considered and the precipitating cloud after 06:00 UTC 08 March is below 2000 m, no hydrometeors are present in the classification of Fig. 12d after 06:00 UTC 08 March.

## 5 Conclusions

In this article we presented a four-months dataset of cloud and precipitation measurements by an X-band polarimetric radar, a W-band Doppler cloud profiler, a multi-angle snowflake camera and a two-dimensional video disdrometer in the PyeongChang region in South Korea during the ICE-POP 2018 campaign. The dataset is unique as it represents, together with other ICE-POP

measurements, the first observations of cloud and precipitation with radars at different frequencies and ground-based in situ measurements in the Taebaek mountains. It is complementary to similar dataset in other regions and allows to compare snowfall microphysical studies in different geographic contexts. In particular, it is relevant to validate conceptual models of orographic precipitation drawn for other mountain chains such as the studies of Houze and Medina (2005); Panziera et al. (2015); Grazioli et al. (2015).

The campaign was characterised by mostly cold, dry and windy weather. However, four major precipitation events took place and contributed to 68% of the total precipitation accumulation over the campaign (25 November 2017 to 15 March 2018). We presented the meteorological conditions and data from these four events. The event with the largest precipitation accumulation (i.e. 28 February 2018) was characterised by an upper-level cyclonic enhancement due to the presence of a PV streamer, which led to a mature frontal system and intense precipitation. This event is further described in Gehring et al. (2020b) and shows

the relevance of this dataset to study microphysics and dynamics of snowfall. The dominant hydrometeor types during the campaign were aggregates and rimed particles. The presence of SLW was confirmed for all events by the presence of graupel particles in MASC images and a hydrometeor classification based on MXPol polarimetric variables. This dataset is particularly suited to study snowfall microphysics, thanks to the synergy between dual-polarisation and spectral information at different frequencies, as well as snowflake photographs.

Future studies could use the data presented in this paper together with other measurements from ICE-POP 2018. This includes radar data at X, Ku and Ka band and is particularly suited for microphysical studies with multi-frequency measurements.

## 6 Data availability

The dataset presented in this paper is available at the PANGAEA platform (https://doi.pangaea.de/10.1594/PANGAEA.918315, Gehring et al. (2020a)). Only the moments of the radar spectra were uploaded, the full spectra can be requested from the

corresponding author.

*Author contributions.*   JG, AB, KA, GL designed the experiment. JG, AB and AF operated the instruments. JG processed and analysed the observational data. AF calibrated the differential reflectivity measurements. ACBR developed the anti-aliasing algorithm. NB computed the radar-based hydrometeor classification. JG, with contributions of all authors, prepared the manuscript.

*Competing interests.*   The authors declare that they have no conflict of interest.

*Acknowledgements.* The authors are greatly appreciative to the participants of the World Weather Research Programme Research Development Project and Forecast Demonstration Project, International Collaborative Experiments for Pyeongchang 2018 Olympic and Paralympic winter games (ICE-POP 2018), hosted by the Korea Meteorological Administration. This work was funded by the Korea Meteorological Administration Research and Development Program under Grant KMI2018-06810. J. Gehring and A. Ferrone acknowledge the financial support from the Swiss National Science Foundation (grant 200020-175700/1). We would like to thank Christophe Praz and Jacques Grandjean for their help in the deployment of the instruments. We are grateful to Kwonil Kim, Geunsu Lyu, Sun-yeong Moon, Hong-Mok Park, Hee-Chul Park, Wonbae Bang, SeungWoo Baek, Kyuhee Shin, Daejin Yeom, Bo-Young Ye, DaeHyung Lee, Choeng-lyong Lee, Eunbi Jeong, Su-jeong Cho for their contribution to the instruments operation and removal. We would finally like to thank Dr. SangWon Joo, Mr. YongHee Lee and Dr. Dong-Kyu Lee from KMA.

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

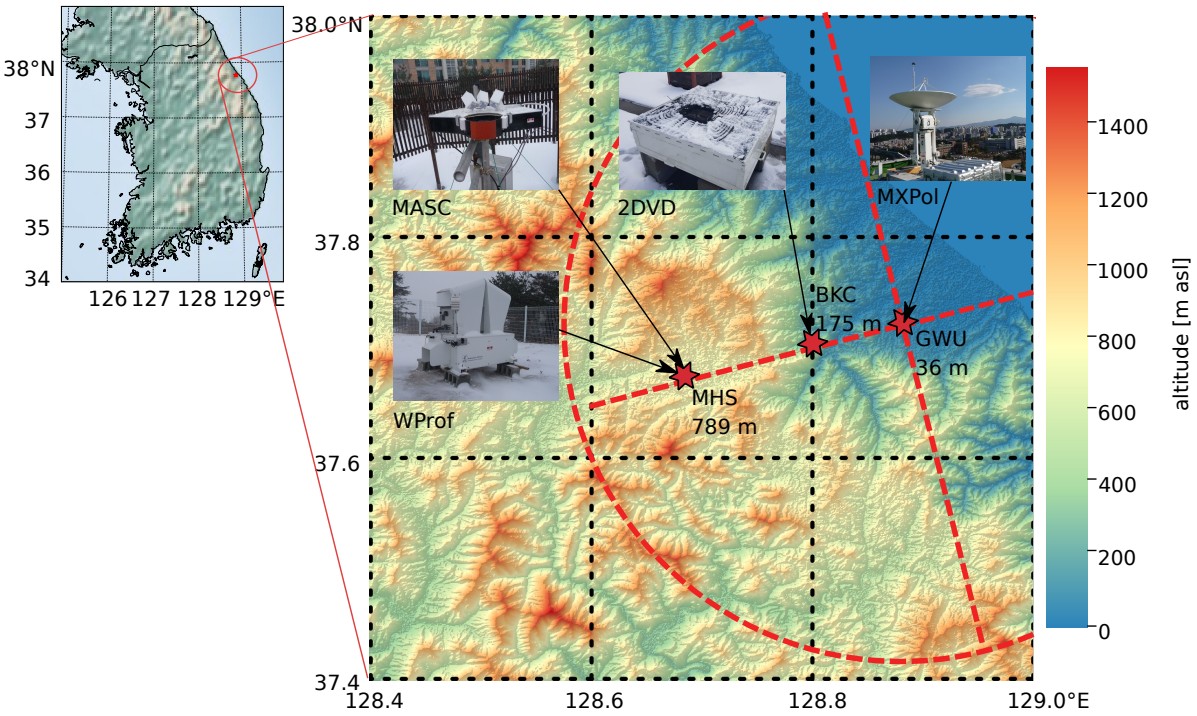

**Figure 1.** Location of the instruments used for this dataset. A digital elevation model shows the topography of the region and its location within South Korea. The red dotted lines and circle show the extent of the main RHIs (27.2 km) and PPI (28.4 km radius) respectively. Note that the MASC was located at BKC from 15 November 2017 to 20 February 2018 and at MHS afterwards. Adapted from Gehring et al. (2020b).

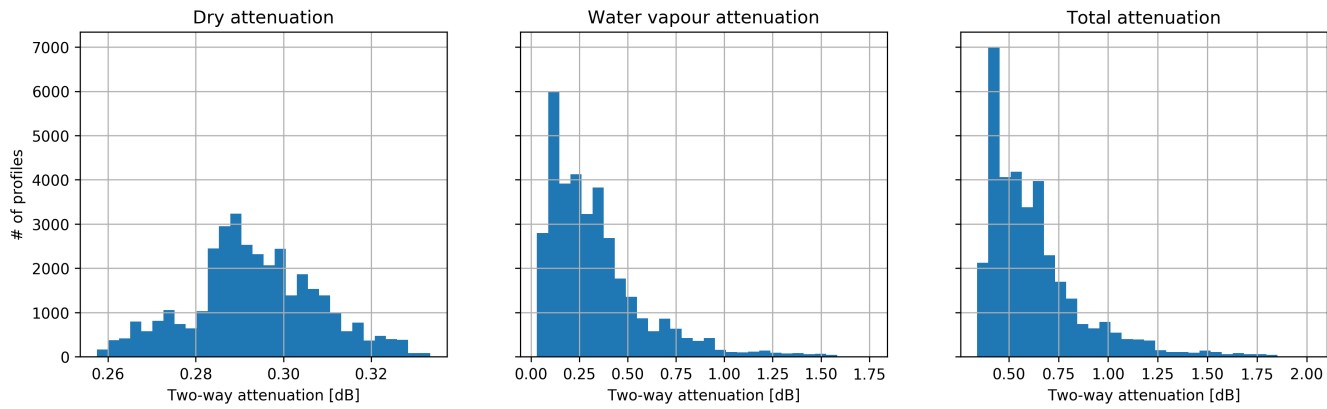

**Figure 2.** Distribution of two-way attenuation (up to 10 km) at 94 GHz for dry air (left) water vapour (centre) and total attenuation (right) computed with PAMTRA for all profiles of WProf interpolating at 5 min the radiosounding data.

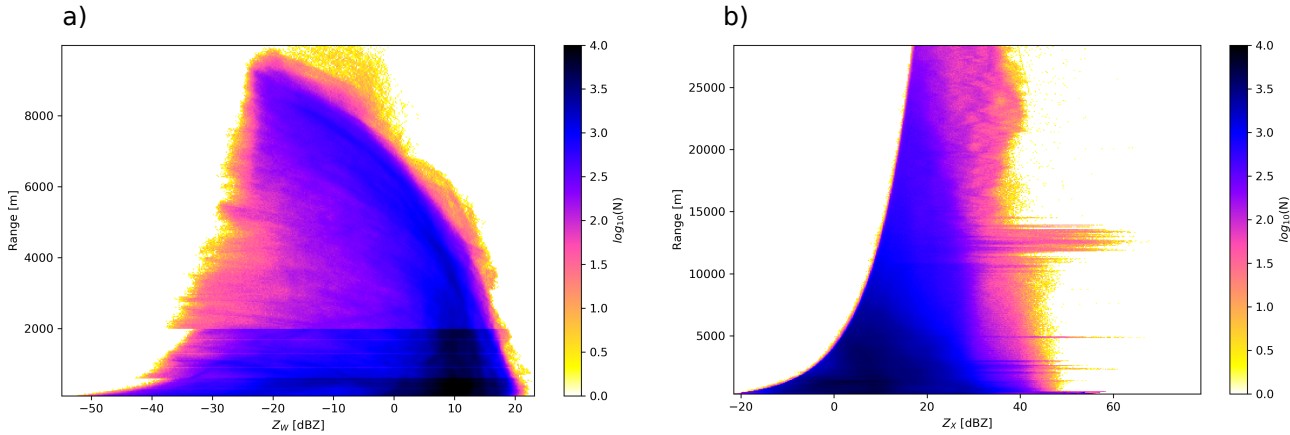

**Figure 3.** Range distribution of reflectivity values for (a) WProf and (b) MXPol during all precipitation events (see Fig. 4). The colour bar shows the number of measurements per range gate. The total number of measurement points is $1.24 \cdot 10^{8}$ for WProf and $2.25 \cdot 10^{8}$ for MXPol.

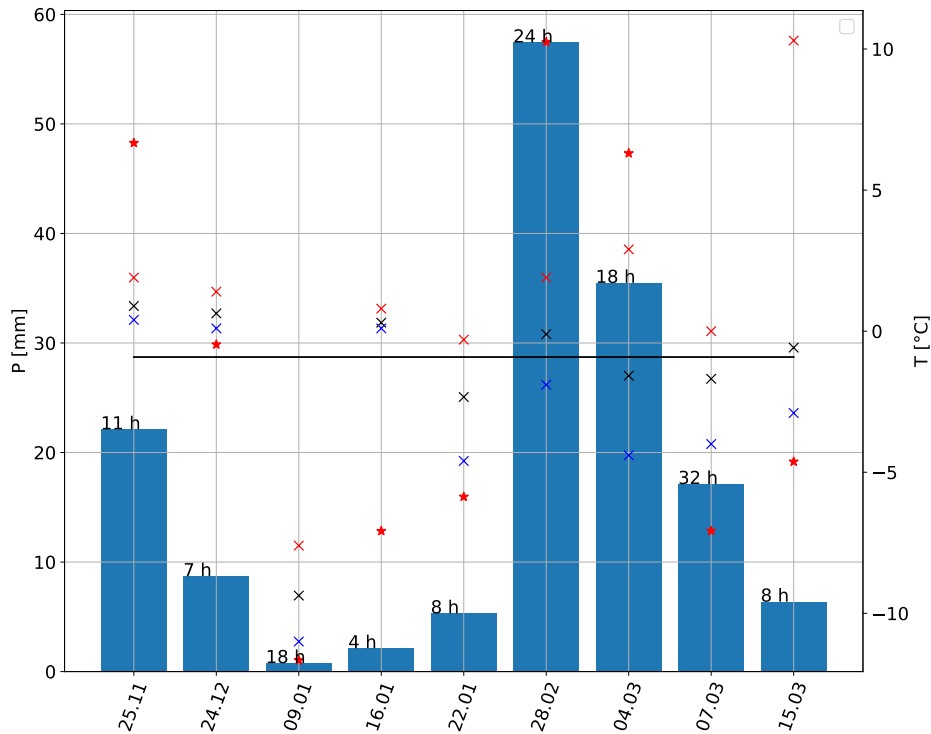

**Figure 4.** Precipitation accumulation (blue bars), mean precipitation rates (red star) in $\mathrm{mm\,h^{-1}}$, maximum temperature (red cross), mean temperature (black cross) and minimum temperature (blue cross). The black line shows the mean precipitation rate during precipitation events. The duration of the events is written on top of the bars. The precipitation and temperature data come for a Pluvio$^2$ weighing rain gauge and a Vaisala weather station located in MHS. The Pluvio$^2$ data are not part of this dataset, but can be requested from the contact author.

25 Nov 2017, 12:00 UTC

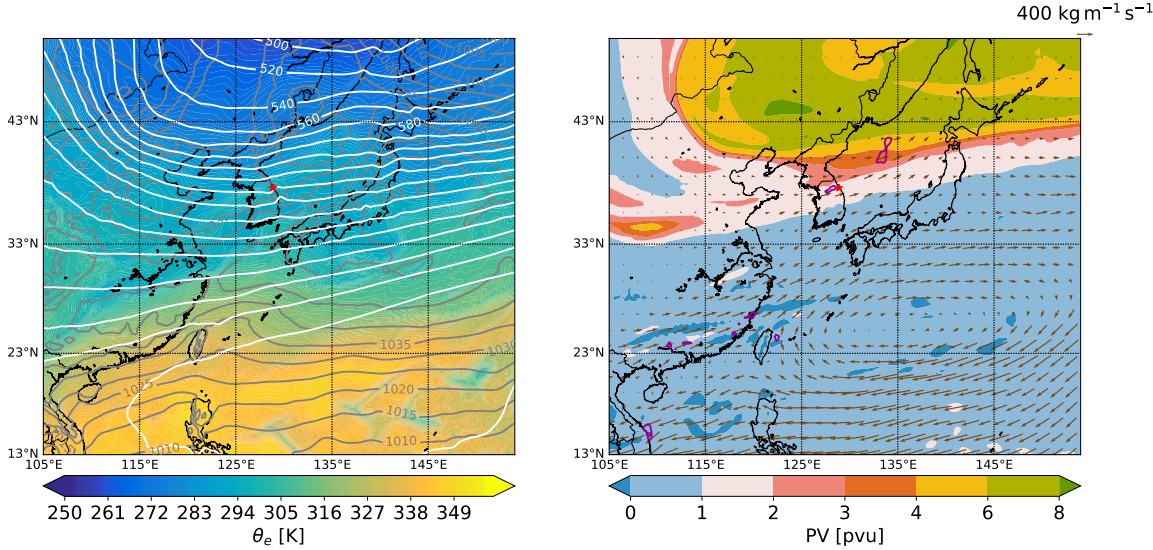

**Figure 5.** Synoptic meteorological fields on 25 November 2017, 12:00 UTC from ERA5 reanalysis. (a) Sea level pressure (grey contours, labels in hPa), 500 hPa geopotential height (white contours, labels in decameters) and 850 hPa equivalent potential temperature ($\theta_e$ colours). (b) Potential vorticity at 315 K (colours), vertically integrated water vapour flux (brown arrows) and precipitation rate (purple contours every $2\,\mathrm{mm\,h^{-1}}$)

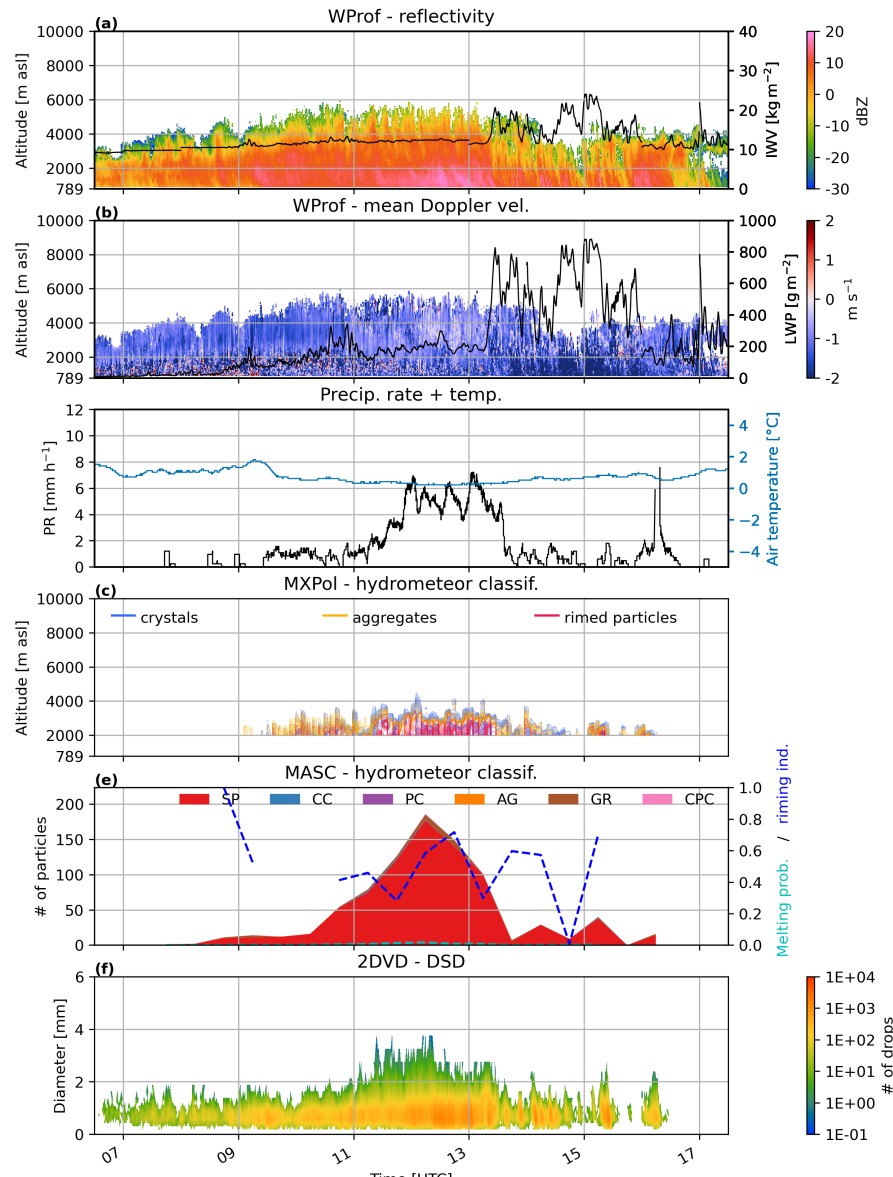

**Figure 6.** Time series on 25 November 2017 of (a) reflectivity and IWV, (b) mean Doppler velocity (defined positive upwards) and LWP from WProf, (c) precipitation rate and air temperature at MHS and (d) hydrometeor classification based on MXPol RHIs towards MHS (averaged between 7 and 20 km). Only data with an elevation angle between 5° and 45° are considered. The isolines represent the proportion of each hydrometeor class normalised by the average number of pixels per time step. The contour interval is 2 %. The blue contours represent crystals, the yellow ones aggregates and the red ones rimed particles. The results are shown only above 2000 m since the lower altitudes are contaminated by ground echoes. (e) Hydrometor classification from the MASC and (f) drop size distribution from the 2DVD. For this event, the MASC was located at BKC at 175 m.a.s.l. together with the 2DVD. The gap around 16:00 UTC of precipitation rate in (c) is due to a technical issue with the rain gauge.

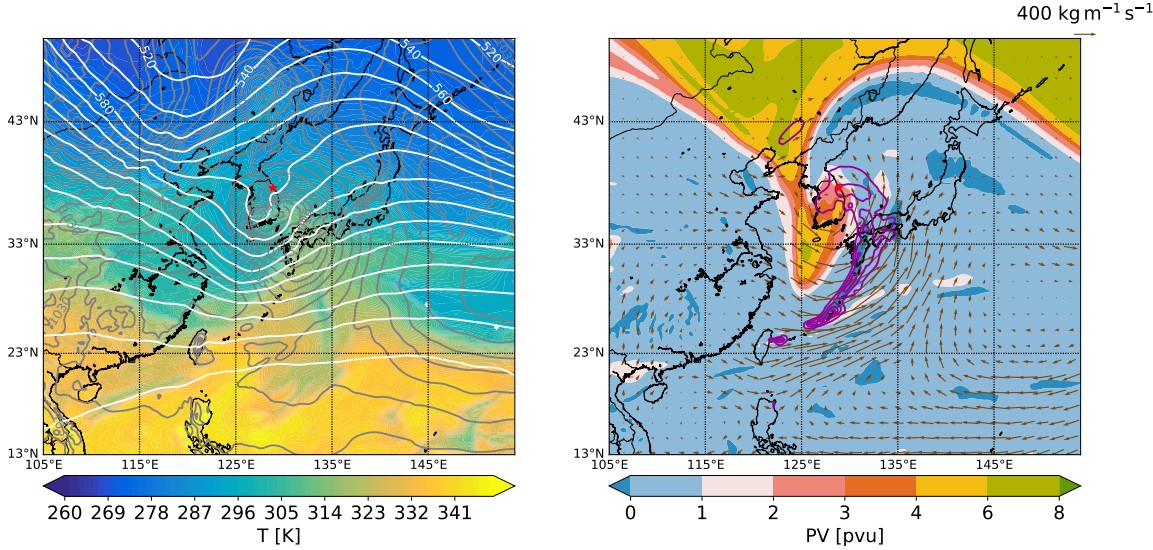

**Figure 7.** Synoptic meteorological fields on 28 February 2018, 12:00 UTC from ERA5 reanalysis. (a) Sea level pressure (grey contours, labels in hPa), 500 hPa geopotential height (white contours, labels in dam) and 850 hPa equivalent potential temperature ($\theta_e$ colours). (b) Potential vorticity at 315 K (colours), vertically integrated water vapour flux (brown arrows) and precipitation rate (purple contours every 2 $\text{mm h}^{-1}$)

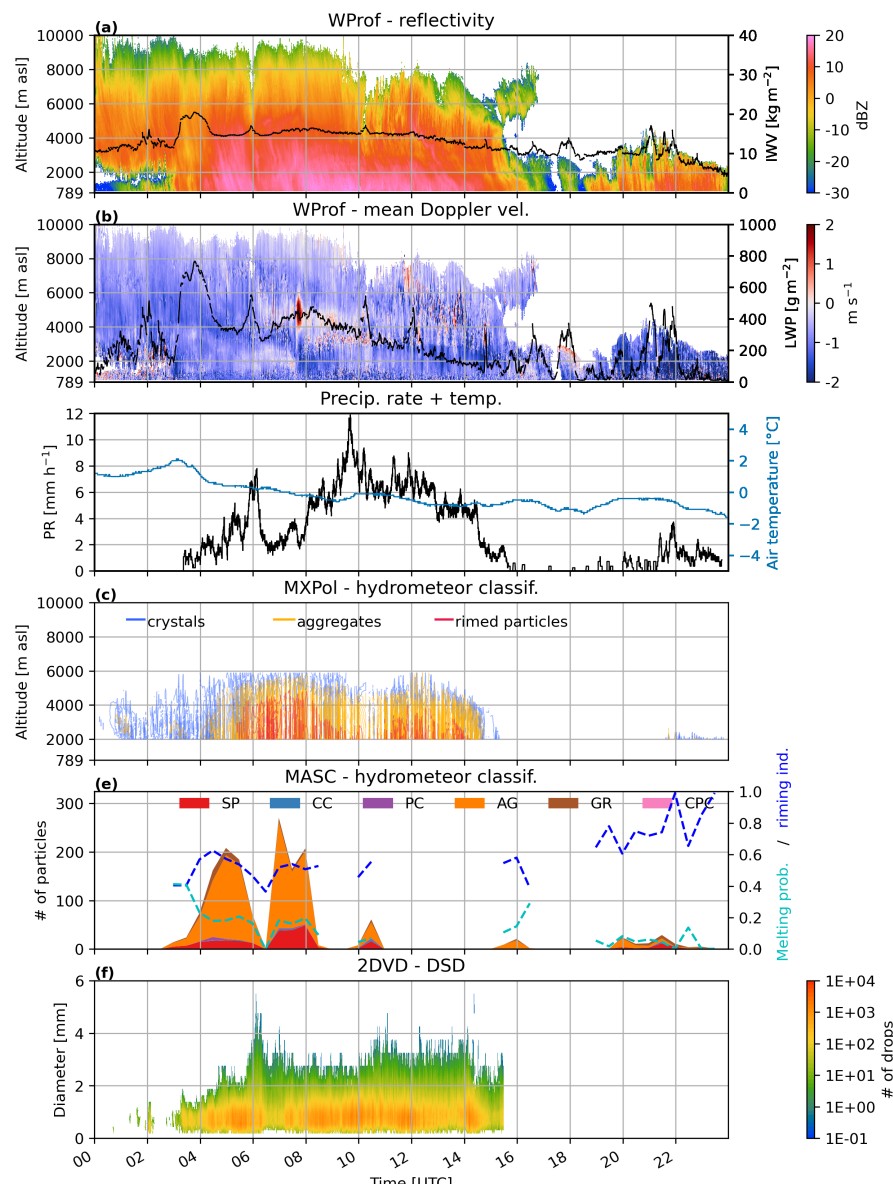

**Figure 8.** Time series on 28 February 2018 of (a) reflectivity and IWV, (b) mean Doppler velocity (defined positive upwards) and LWP from WProf, (c) precipitation rate and air temperature at MHS and (d) hydrometeor classification based on MXPol RHIs towards MHS (averaged between 7 and 20 km). Only data with an elevation angle between 5° and 45° are considered. The isolines represent the proportion of each hydrometeor class normalised by the average number of pixels per time step. The contour interval is 2 %. The blue contours represent crystals, the yellow ones aggregates and the red ones rimed particles. The results are shown only above 2000 m since the lower altitudes are contaminated by ground echoes. (e) Hydrometor classification from the MASC and (f) drop size distribution from the 2DVD. Note that the MASC was located at MHS at 789 m a.s.l. as WProf, while the 2DVD was located at BKC at 175 m a.s.l.

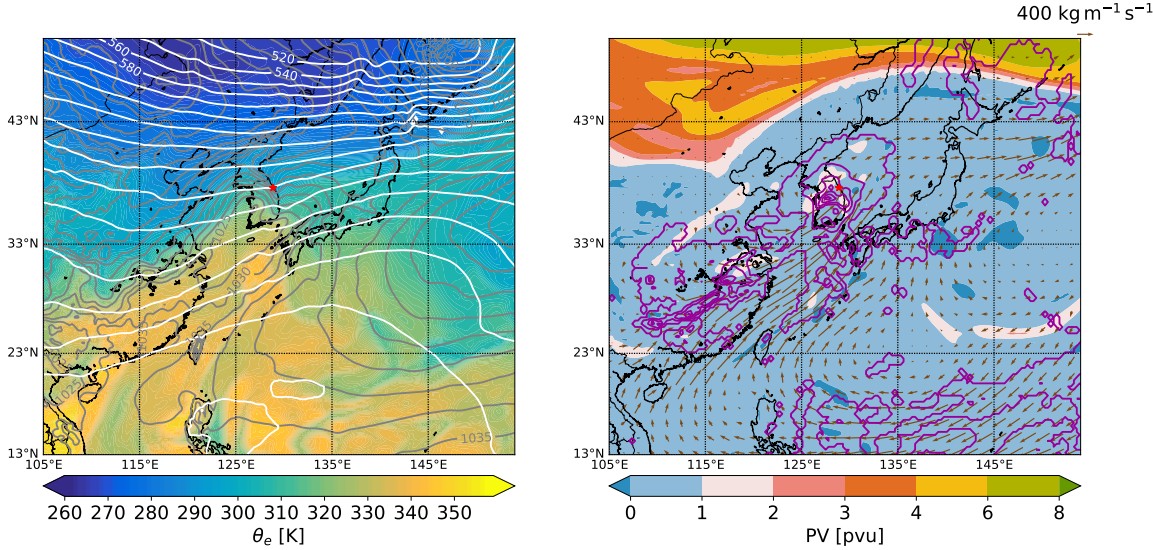

**Figure 9.** Synoptic meteorological fields on 04 March 2018, 15:00 UTC from ERA5 reanalysis. (a) Sea level pressure (grey contours, labels in hPa), 500 hPa geopotential height (white contours, labels in dam) and 850 hPa equivalent potential temperature ($\theta_e$ colours). (b) Potential vorticity at 315 K (colours), vertically integrated water vapour flux (brown arrows) and precipitation rate (purple contours every $2\,\mathrm{mm\,h^{-1}}$)

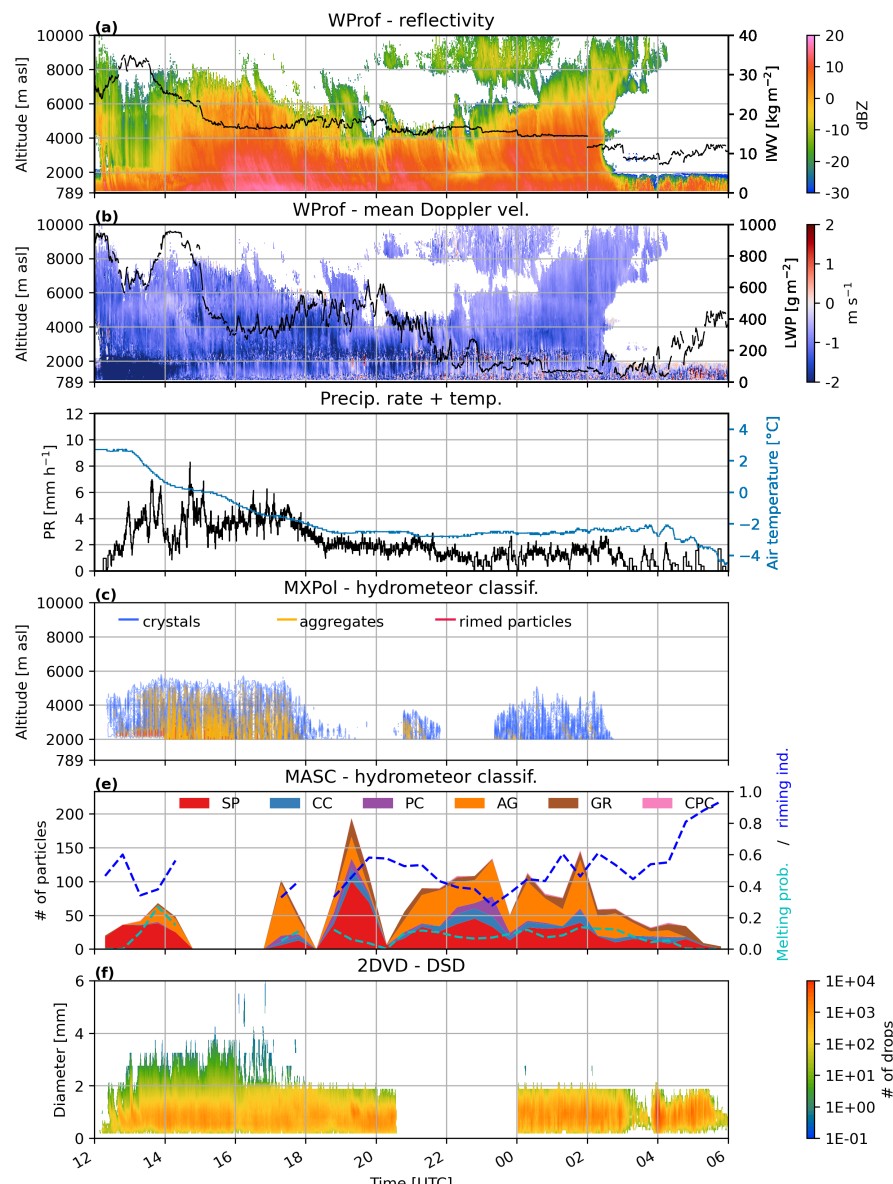

**Figure 10.** Time series from 04 to 05 March 2018 of (a) reflectivity and IWV, (b) mean Doppler velocity (defined positive upwards) and LWP from WProf, (c) precipitation rate and air temperature at MHS and (d) hydrometeor classification based on MXPol RHIs towards MHS (averaged between 7 and 20 km). Only data with an elevation angle between 5° and 45° are considered. The isolines represent the proportion of each hydrometeor class normalised by the average number of pixels per time step. The contour interval is 2 %. The blue contours represent crystals, the yellow ones aggregates and the red ones rimed particles. The results are shown only above 2000 m since the lower altitudes are contaminated by ground echoes. (e) Hydrometor classification from the MASC and (f) drop size distribution from the 2DVD. Note that the MASC was located at MHS at 789 m a.s.l. as WProf, while the 2DVD was located at BKC at 175 m a.s.l.

08 Mar 2018, 00:00 UTC

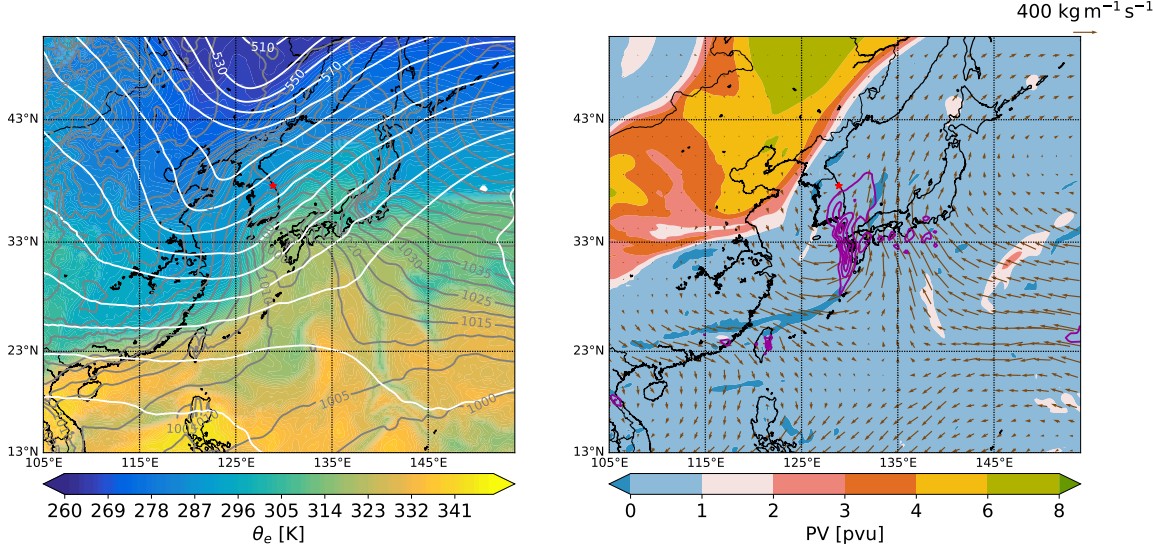

**Figure 11.** Synoptic meteorological fields on 08 March 2018, 00:00 UTC from ERA5 reanalysis. (a) Sea level pressure (grey contours, labels in hPa), 500 hPa geopotential height (white contours, labels in dam) and 850 hPa equivalent potential temperature ($\theta_e$ colours). (b) Potential vorticity at 315 K (colours), vertically integrated water vapour flux (brown arrows) and precipitation rate (purple contours every 2 mm h$^{-1}$)

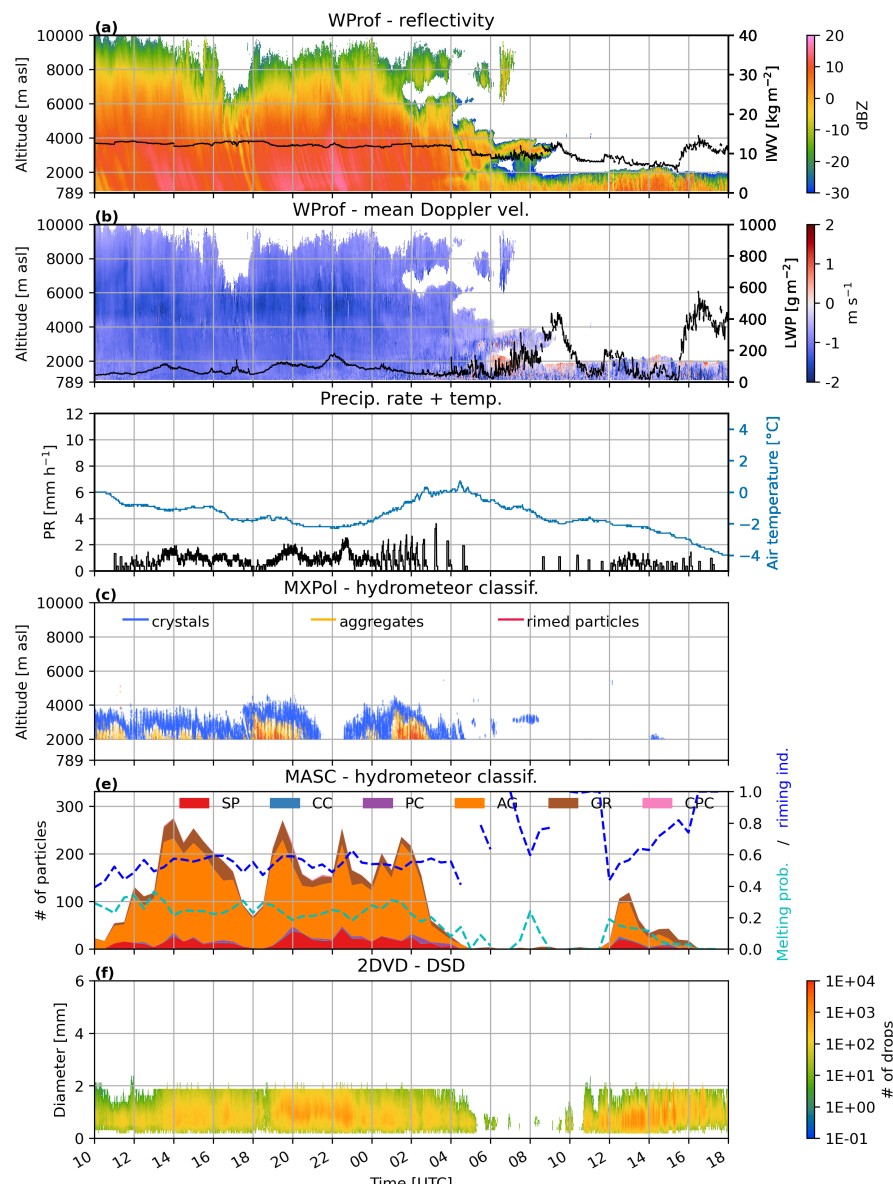

**Figure 12.** Time series from 07 to 08 March 2018 of (a) reflectivity and IWV, (b) mean Doppler velocity (defined positive upwards) and LWP from WProf, (c) precipitation rate and air temperature at MHS and (d) hydrometeor classification based on MXPol RHIs towards MHS (averaged between 7 and 20 km). Only data with an elevation angle between 5° and 45° are considered. The isolines represent the proportion of each hydrometeor class normalised by the average number of pixels per time step. The contour interval is 2 %. The blue contours represent crystals, the yellow ones aggregates and the red ones rimed particles. The results are shown only above 2000 m since the lower altitudes are contaminated by ground echoes. (e) Hydrometor classification from the MASC and (f) drop size distribution from the 2DVD. Note that the MASC was located at MHS at 789 m a.s.l. as WProf, while the 2DVD was located at BKC at 175 m a.s.l.

**Table 1.** Description of WProf chirps

|  | Range | Range resolution | Doppler interval | Doppler resolution | Integration time |
|---|---|---|---|---|---|
| **chirp2** | [2016, 9984] m | 32.5 m | [-5.1, 5.1] $\mathrm{m\,s^{-1}}$ | 0.020 $\mathrm{m\,s^{-1}}$ | 0.82 s |
| **chirp1** | [603, 1990] m | 11.2 m | [-5.1, 5.1] $\mathrm{m\,s^{-1}}$ | 0.020 $\mathrm{m\,s^{-1}}$ | 0.37 s |
| **chirp0** | [100, 598] m | 5.6 m | [-7.16, 7.13] $\mathrm{m\,s^{-1}}$ | 0.028 $\mathrm{m\,s^{-1}}$ | 0.18 s |

**Table 2.** Specifications of MXPol and WProf

| Specifications | MXPol | WProf |
|---|---|---|
| Frequency | 9.41 GHz | 94 GHz |
| 3 dB beamwidth | 1.27° | 0.53° |
| Sensitivity at 8 km | 5 dBZ | -40 dBZ |
| Transmission type | pulsed | FMCW |
| Polarisation | dual-polarisation | single-polarisation |
| Range resolution | 75 m | 5.6, 11.2, 32.5 m |

**Table 3.** Date and time of the start and end of the precipitation events. The four majour events presented here are highlighted in bold.

| ID | Start [UTC] | End [UTC] |
|----|-------------|-----------|
| **1** | **06:30 25 Nov 2017** | **17:30** |
| 2 | 00:00 24 Dec 2017 | 07:00 |
| 3 | 00:00 09 Jan 2018 | 18:00 |
| 4 | 20:00 16 Jan 2018 | 23:59 |
| 5 | 06:30 22 Jan 2018 | 14:30 |
| **6** | **00:00 28 Feb 2018** | **23:59** |
| **7** | **12:00 04 Mar 2018** | **06:00 05 Mar 2018** |
| **8** | **10:00 07 Mar 2018** | **18:00 08 Mar 2018** |

**Table 4.** Data amount for all instruments

| | |
|---|---|
| MXPol | 62 h, 4166 RHIs, 2036 PPIs |
| WProf | 121 h, 146'548 profiles |
| MASC | 29'886 triplets |
| 2DVD | 2'304'730 drops |