# Peer review of "Radar and ground-level measurements of precipitation collected by EPFL during the ICE-POP 2018 campaign in South-Korea"

_Earth System Science Data, 2020_

## Referee Comment (RC1) · Anonymous Referee #1 · 17 Aug 2020

This manuscript describes selected snowfall cases from a four-months dataset of precipitation and cloud measurements collected in South Korea in 2018. The dataset includes polarimetric radar data from a scanning dual-pol X-Band Doppler radar and a vertically pointing W-Band FMCW radar/radiometer. Besides hydrometeor classifications from the X-Band polarimetry, the dataset also contains information about PSD and habits from two optical disdrometers (2DVD and MASC). As combined radar+in-situ datasets of snowfall are still rare (especially from this region), I think that this dataset will be valuable for microphysical studies and model evaluation. Overall I find the paper nicely written and it certainly matches the scope of ESSD. I think some more details need to be added and described for some of the instruments and procedures,

which I think could help the future data user to work with this dataset.

Introduction: I understand that the focus of this paper is on Korea but due to rareness of combined in-situ and remote sensing snowfall measurements, the authors might want to think about including a list of available datasets from other regions/campaigns at this point. I think it would put this dataset in a larger perspective and provide the reader some idea what datasets are available. Some examples, which come to my mind would be: Hyytiälä (Finland) dataset (BAECC campaign), Barrow/Oliktok ARM sites, ARM AWARE campaign in Antarctica as well as the Antarctic campaigns of your own group, TOSCA campaign in the German Alps, Olympex campaign in Cascade mountains, Long-term observations ICECAPS at summit station Greenland. (I included some references at the end)

P.2, L.31: For non-radar expert readers: Can you explain whether there are other differences of the radar data obtained by DPP and FFT mode except the different Nyquist range?

P.3, L. 3: add comma before "respectively"

Table 1: The heading "Integration time" is not on top of the right column

P.3 L.12 ff, Description of WProf: I think here you should add much more details and additional information, which can become important once somebody aims to work with the data: a) Does your system provide LDR? b) Unfortunately, several of those new W-band radar systems have issues with "ghost echos" (mirror signals when there are strong reflecting targets). It seems to me from the plots that you are lucky and you don't have those problems? Or did you remove them? If so, how? c) You mention that the calibration was done by the manufacturer. Did that include calibration with external targets (sphere or rainfall and Ze calculated from collocated disdrometer) or was it only the calibration of the internal components? How accurate does RPG estimate the calibration to be? d) How was the radar pointing evaluated? e) Did you calibrate the passive channel with liquid nitrogen before/during the campaign? f) Did you change

the radomes between the calibration at RPG and start of your campaign? This might change the calibration. Unfortunately, the radome coating also quickly deteriorates, which can cause several dB of attenuation due imperfect removal of drops. So it would be important to know if they were the same during the entire campaign and in what shape. Did you start the measurements with new or old radomes? Did you run the radar with the strong blower switched on all the time? g) Regarding the LWP: I agree, with the (calibrated!) single passive channel you can infer LWP but only if you can constrain integrated water vapour. How was that done in your case? The liquid water path is actually an interesting variable in order to estimate total path integrated attenuation but also for riming.

Also some technical notes: I suggest to mention the Table 1 already after you mention the chirps (L. 14). Küchler et al., not only describes the passive part but the entire W-Band FMCW system, so you might consider mention it earlier, then you can also avoid to mention it again at the end of the paragraph.

P.6, Gas attenuation correction: This description is not entirely clear to me. I understand that you calculate attenuation due to the main attenuating gases at W-Band, which are water vapour and oxygen. I understand that you use the radio sonde profiles to estimate the gas attenuation profile but are you using a constant profile for all the times between two launches or do you interpolate the RS profiles in between? I didn't check the data files but I suggest to include all correction, such as gas attenuation profiles as additional arrays if the raw reflectivity profiles are not provided. It allows the user to retrieve the radar profiles without any of your corrections applied and so the user can maybe also apply his own corrections. The comparison with Dias Neto et al. makes only sense if the columnar water vapour amounts had similar range for your campaign. Did they?

P.6, L. 16: Correct citation is "Dias Neto et al., 2019"

Fig. 6 and similar figures for the other cases: The choice of color table is certainly

always a matter of taste but I am wondering whether you considered to use a color table for reflectivity and the diameters which contains more than the three colors that you have now. I could image this could show much more structural information. Maybe you can also adjust/reduce the Doppler velocity limits to show better the slight changes from unrimed to rimed snow, which I would guess are often only between 1 and 2.5 m/s. If you extract ZDR from the X-Pol, did you consider to calculate dual-wavelength ratios between X and W-Band? It might be quite noisy due to the strongly different volumes but in stratiform snow it might still provide some useful information.

Caption Fig 4: Full Stop missing at the end of caption.

P.6, L27: Space missing after comma and before "2 km"

Table 3: The ":" separating hours and minutes is missing on several entries.

Fig. 5: Write out that "dam" means decameters, it might not be a familiar "unit" to all readers.

Fig.6: Again, in the current color scale of the W-band, the variations of mean Doppler velocity, which one could expect due to riming are extremely hard to see. I suggest to experiment more with other color scales in order to better visualize those structures.

Fig. 6: It might be worth mentioning in the caption that the 2DVD was measuring at a different location and much lower altitude. This also avoids confusion that the 2DVD was really only measuring rain and not snow (although it could measure snow as well). In this current multi-panel plot it gives the impression all measurements would be from the same location.

P 7, L. 23: Is the degree of riming set to zero for the rain cases in the data? Maybe this would be good to avoid confusion for users. At least it should be mentioned in the meta data if not already included.

P. 8, L.1: Maybe better "At 00:00 UTC on 28 February, the nimbostratus. . ."

P. 8, L. 2: How do you know that this is a fog layer and not for example a thin mixed-phase cloud, or drizzle cloud?

Fig. 6 and following radar multi-panel plots: I would find it interesting to have temperature isotherms overplotted in one of the graphs. If you don't have this information, then maybe juts plot the surface temperature. I find it quite difficult to interpret the data without this essential variable.

Fig. 8: I am surprised that the mean Doppler velocities in the W-Band are so similar during periods when the MXPol classifies predominantly aggregates and rimed particles. Shouldn't the vertically pointing Doppler velocities be much faster if they are really rimed? At which rime mass fraction does the MXPol classification detect them as rimed? Isn't it also surprising that the MASC derived rime mass fraction is not really correlating with the MXPol classification? Are you able to provide an error bar for the rime mass fraction?

Fig. 10: I think you should somehow clearly mark time periods with rain versus others with snow. Especially since the MASC is deriving rime mass fraction for rain drops were it should be set to zero. It would be good if this correction/flagging of the MASC data for rain is done by you in order to avoid somebody is using it in a wrong way.

P. 10, L. 4: Change K-u, K-a into common "Ku" and "Ka"

References:

Lubin D, D Zhang, I Silber, R Scott, P Kalogeras, A Battaglia, D Bromwich, M Cadeddu, E Eloranta, A Fridlind, A Frossard, K Hines, S Kneifel, W Leaitch, W Lin, J Nicolas, H Powers, P Quinn, P Rowe, L Russell, S Sharma, J Verlinde, and A Vogelmann. 2020. "AWARE: The Atmospheric Radiation Measurement (ARM) West Antarctic Radiation Experiment." Bulletin of the American Meteorological Society, 101(7), 10.1175/BAMS-D-18-0278.1.

Petäjä, T., and Coauthors, 2016: BAECC: A Field Campaign to Elucidate the Impact of

Biogenic Aerosols on Clouds and Climate. Bull. Amer. Meteor. Soc., 97, 1909–1928, https://doi.org/10.1175/BAMS-D-14-00199.1.

Löhnert, U., S. Kneifel, A. Battaglia, M. Hagen, L. Hirsch, and S. Crewell, 2011: A Multisensor Approach Toward a Better Understanding of Snowfall Microphysics: The TOSCA Project. Bull. Amer. Meteor. Soc., 92, 613–628, https://doi.org/10.1175/2010BAMS2909.1.

Houze, R. A., and Coauthors, 2017: The Olympic Mountains Experiment (OLYMPEX). Bull. Amer. Meteor. Soc., 98, 2167–2188, https://doi.org/10.1175/BAMS-D-16-0182.1.

Shupe, M. D., and Coauthors, 2013: High and Dry: New Observations of Tropospheric and Cloud Properties above the Greenland Ice Sheet. Bull. Amer. Meteor. Soc., 94, 169–186, https://doi.org/10.1175/BAMS-D-11-00249.1.

---

## Referee Comment (RC2) · Anonymous Referee #2 · 27 Nov 2020

The paper titled "Radar and ground-level measurements of precipitation collected by EPFL during the ICE-POP 2018 campaign in South-Korea by Gehring et al" describes the four-month dataset of precipitation and cloud measurements collected during ICE-PO 2018. While I believe the paper is well written, I am not sure if the authors made an attempt to convince a non-expert in this subject as to how important this data set is to the scientific community. I also see that the author is already published an article in ACP partly in regard to this data set focusing on one particular event of 28 February 2018. May be including a little bit more on other available data sets in this category and why these measurements are important compared to other measurements may help the readers who are not experts in this subject. While this dataset seem to be unique,

I believe the authors can do a better job in analyzing the available data and may be comparing with other available measurements. Otherwise, this is just another data set. I, therefore recommend authors to do further analyses to show the uniqueness of the data set and it would be helpful for the readers if the authors could explain this data set a bit more with more analyses such as a relationship between precipitation and reflectivity for both W-band and X-band radars.

Here are my comments:

Line 6, page 1 : please use a period following "video disdrometer (https://doi.pangaea.de/10.1594/PANGAEA.918315, Gehring et al. (2020a))"

Line 15, page 2 : It may be helpful to add something about how unique your measurements are and the importance of the data to the scientific community particularly to microphysics and their use in evaluating models.

Line 23, page 2: Isn't this figure same as figure 1 of Gehring et al., 2020 paper?

Line 1, page 7: The plots you show for these events (25 November 2017, 28 February 2018, 04 and 07 March 2018) are just a description of what you see in your data. It would be helpful if you could explain this data set a bit more with more analyses such as some sort of relationship between precipitation and reflectivity for both W-band and X-band radars and may be comparing the reflectivity with some other precipitation measurements that are available. I think this would add significance of this data set and the paper.

Line 15, page 7: Since this is the third-largest precipitation accumulation in the record, it may be helpful to see a precipitation rate time series plot and temperature if possible. May be if possible it will be appropriate to do the same sort of analyses for all the main events listed here.

Line 26-32, page 8: I think it may be appropriate to do further analyses here as well as commented above.

Once additional analyses are done, I would revise the "conclusions" section a little bit to include the uniqueness of the measurements and how they compare with other measurements (if they are available).Some sort of validation of the data may help strengthen the paper and the robustness of the data set I believe.

---

## Author Comment (AC1) · 10 Dec 2020

**Response to reviews, ESSD manuscript ESSD-2020-134 "Radar and ground-level measurements of precipitation collected by EPFL during the ICE-POP 2018 campaign in South-Korea" by Gehring et al.**

**Dear ESSD Editor,**

Please find in this document our answers to the referees' comments. We hope that our corrections to the manuscript will make it suitable for publication in ESSD.

Yours sincerely,

The authors of "Radar and ground-level measurements of precipitation collected by EPFL during the ICE-POP 2018 campaign in South-Korea"

**Response to the referees' comments:**

We thank both reviewers for their constructive comments on the manuscript. We improved and added significant information in the dataset, which can be summarised as follows :

- We computed the liquid water path (LWP) and the integrated water vapour (IWV) with measurements from the 89 GHz radiometer following the method of Billault-Roux and Berne, 2020. This includes a bias correction of the brightness temperature of 20 K.
- We computed the attenuation from atmospheric gases of WProf reflectivity measurements interpolating the three-hourly radiosoundings data and using the Passive and Active MicrowaveTRansfer Model (PAMTRA, Mech et al., 2020). We now provide the reflectivity measurements with and without a correction for atmospheric gas attenuation.
- We corrected a calibration issue in WProf measurements that amounts to 6.3 dB. This is clearly stated in the manuscript and in the metadata. Note that this changes the values of sensitivity interpreted from Fig. 3.
- We added temperature, melting information and precipitation rate on the figures summarising each event.

**Anonymous Referee #1**

Received and published: 17 August 2020

1. This manuscript describes selected snowfall cases from a four-months dataset of precipitation and cloud measurements collected in South Korea in 2018. The dataset includes polarimetric radar data from a scanning dual-pol X-Band Doppler radar and a vertically pointing W-Band FMCW radar/radiometer. Besides hydrometeor classifications from the X-Band polarimetry, the dataset also contains information about PSD and habits from two optical disdrometers (2DVD and MASC). As combined radar+in- situ datasets of snowfall are still rare (especially from this region), I think that this dataset will be valuable for microphysical studies and model evaluation. Overall I find the paper nicely written and it certainly matches

the scope of ESSD. I think some more details need to be added and described for some of the instruments and procedures, which I think could help the future data user to work with this dataset.

2. Introduction: I understand that the focus of this paper is on Korea but due to rareness of combined in-situ and remote sensing snowfall measurements, the authors might want to think about including a list of available datasets from other regions/campaigns at this point. I think it would put this dataset in a larger perspective and provide the reader some idea what datasets are available. Some examples, which come to my mind would be: Hyytiälä (Finland) dataset (BAECC campaign), Barrow/Oliktok ARM sites, ARM AWARE campaign in Antarctica as well as the Antarctic campaigns of your own group, TOSCA campaign in the German Alps, Olympex campaign in Cascade mountains, Long-term observations ICECAPS at summit station Greenland. (I included some references at the end)

Thank you for this suggestion. We added references to these field campaigns in the introduction on p.2 L. 4-25:

"Several past field campaigns demonstrated the usefulness of combined remote sensing and in situ measurements for snowfall studies. The TOSCA project in the Bavarian Alps in Germany (Löhnert et al., 2011) combined vertically pointing radars, radiometers and optical disdrometers among others, to better characterise the vertical distribution of snowfall for satellite retrievals and numerical model validations. During the 2015/16 fall-winter season, the OLYMPEX campaign (Houze et al., 2017) took place in the vicinity of the mountainous Olympic Peninsula, USA, to study how Pacific precipitation systems are influenced by the orography. The BAECC field campaign (Petäjä et al., 2016) provided eight months of measurements in Hyytiälä, Finland, to study biogenic aerosols, clouds, and precipitation and their interactions. In-situ and remote sensing instruments have also proven very useful to study atmospheric radiation, cloud and precipitation properties in polar regions. The North Slope of Alaska atmospheric observatory (Verlinde et al., 2016) in Barrow and Oliktok provides a long series of measurements including radiometers, lidars, cloud radars and a MASC among many other instruments. Another major Arctic measurement site is located at Summit, Greenland, where the ICECAPS (Shupe et al., 2013) field campaign was conducted to collect measurements of radiation, clouds and precipitation to study the energy and hydrological budgets of the Greenland Ice Sheet. In Antarctica, the APRES3 field campaign (Genthon et al., 2018) provided the first dual-polarisation radar measurements from November 2015 to February 2016. Along with snowflake photographs, micro rain radar and lidar measurements, the dataset led to unprecedented insights in Antarctic snowfall microphysics (Grazioli et al., 2017a). Finally, the AWARE campaign (Lubin et al., 2020) gathered cloud radars, lidars, radiometers, aerosols and microphysical measurements from December 2015 to December 2016 in McMurdo Station, Antarctica. The AWARE dataset offers numerous cases for mixed-phase cloud parametrisation in weather and climate models. The measurements of these field campaigns allowed for innovative studies and new insights in cloud and precipitation processes in these various regions (Kalesse et al., 2016; von Lerber et al., 2017; Grazioli et al., 2017b; Cole et al., 2017; Zagrodnik et al., 2019). For a better understanding of cloud and precipitation processes in the Taebaek mountains, a field campaign combining remote sensing and in situ measurements is needed."

3. P.2, L.31: For non-radar expert readers: Can you explain whether there are other differences of the radar data obtained by DPP and FFT mode except the different Nyquist range?

The main difference is that in FFT mode one can retrieve the full Doppler spectrum. This information was written at the end of the paragraph, but for clarity we changed the organisation of the paragraph to make it more obvious that it is the most important difference between FFT and DPP modes. P.3, L14-18:

"The main variables retrieved from MXPol measurements in dual-pulse pair (DPP) mode are the equivalent reflectivity factor at horizontal polarisation ZH (dBZ), the differential reflectivity ZDR (dB), the specific differential phase shift on propagation Kdp ( $^{\circ}$  km-1), the copolar correlation coefficient phv, the mean Doppler velocity VD (m s-1) and the Doppler spectral width  $\sigma v$  (m s-1). Additionally, in fast-Fourier transform (FFT) mode, the full Doppler spectrum at 0.17 m s-1 resolution is retrieved at each range gate."

4. P.3, L. 3: add comma before "respectively"

We added the comma.

5. Table 1: The heading "Integration time" is not on top of the right column

Thank you for noting that, we corrected it.

6. P.3 L.12 ff, Description of WProf: I think here you should add much more details and additional information, which can become important once somebody aims to work with the data: a) Does your system provide LDR? b) Unfortunately, several of those new W-band radar systems have issues with "ghost echos" (mirror signals when there are strong reflecting targets). It seems to me from the plots that you are lucky and you don't have those problems? Or did you remove them? If so, how? c) You mention that the calibration was done by the manufacturer. Did that include calibration with external targets (sphere or rainfall and Ze calculated from collocated disdrometer) or was it only the calibration of the internal components? How accurate does RPG estimate the calibration to be? d) How was the radar pointing evaluated? e) Did you calibrate the passive channel with liquid nitrogen before/during the campaign? f) Did you change the radomes between the calibration at RPG and start of your campaign? This might change the calibration. Unfortunately, the radome coating also quickly deteriorates, which can cause several dB of attenuation due imperfect removal of drops. So it would be important to know if they were the same during the entire campaign and in what shape. Did you start the measurements with new or old radomes? Did you run the radar with the strong blower switched on all the time? g) Regarding the LWP: I agree, with the (calibrated!) single passive channel you can infer LWP but only if you can constrain integrated water vapour. How was that done in your case? The liquid water path is

actually an interesting variable in order to estimate total path integrated attenuation but also for riming.

7. Also some technical notes: I suggest to mention the Table 1 already after you mention the chirps (L. 14). Küchler et al., not only describes the passive part but the entire W-Band FMCW system, so you might consider mention it earlier, then you can also avoid to mention it again at the end of the paragraph.

a) Since it is a single polarisation radar, it does not provide LDR. We decided not to add this information in the paper, since it is not mentioned that it has dual-polarisation capacity, it seems not necessary to mention if it has LDR or not.

b) We never noticed ghost echoes with this radar, also by looking at spectrograms.

c) Thank you for this question, which made us realise that due to a problem with the radar software version we were using during this campaign the calibration was not correct. After discussion with RPG, we had to add 6.3 dB to the reflectivity values currently published in PANGEA. This includes the correction due to a software issue as well as an end to end calibration with disdrometer which amounts to 0.85 dB  $\pm$  1 dB. So yes the calibration now includes comparison with a disdrometer and the final accuracy is  $\pm$  1 dB. We added this information in the paper and in the metadata. (see below the quotation from the revised manuscript)

d) The radar pointing was evaluated by checking the levels at the beginning and the end of the campaign. The levels shown that the radar was pointing almost perfectly vertically. However, since the vertical alignment was not monitored constantly, the spectral and Doppler velocity data should be interpreted carefully, especially in case of strong horizontal winds.

e) Yes RPG calibrated the radiometer with liquid nitrogen before the campaign. We added this information in the manuscript.

f) The radomes were not changed between the calibration and the campaign and were in good shape. The blowers were switched on all the time. We added this information in the manuscript.

g) The dataset previously submitted contained the LWP from RPG's algorithm. We decided to use a new non-site dependent algorithm that was developed to estimate IWV and LWP from this single channel 89-GHz radiometer (Billault-Roux and Berne 2020). So now both IWV and LWP are available and were computed using three-hourly radiosounding data. We also took the opportunity to correct the bias of the brightness temperature in the WProf files. The explanation of how this bias was computed can be found in Sect. 6.2 of Billault-Roux and Berne, 2020. We added this information in the paper and the metadata. We also added the LWP on the (b) panels of Figs. 6,8,10.

These modifications have been included on P. L.2-17: "WProf contains a 89 GHz radiometer, which can be used to retrieve the liquid water path (LWP) and the integrated water vapour (IWV). We computed LWP and IWV using the method described in Billault-Roux and Berne (2020). The brightness temperature measurements had a bias of 20 K, which we corrected. After correction the root mean squared error (RMSE) is 2.88 K, taking radiosoundings as the reference. The

RMSE of LWP and IWV are 86.5 g m2 and 1.72 kg m2 respectively, including the RMSE of brightness temperature in the input of the algorithm. More information on the uncertainty of this algorithm can be found in Billault-Roux and Berne (2020). In particular, note that the accuracy is deteriorated in case of intense precipitation. WProf was calibrated by the manufacturer Radiometer Physics GmbH just before the ICE-POP 2018 campaign. This included a calibration of the 89 GHz radiometer with liquid nitrogen and a calibration of the radar with disdrometers following the method of Myagkov et al. (2020). The uncertainty of WProf reflectivity calibration is +/- 1 dB. Note that compared to Gehring et al. (2020b) the calibration was updated following RPG's recommendations and hence the Ze values of WProf do not match with those of this article. This calibration correction was applied on the data available on PANGEA (Gehring et al., 2020a). The radomes were in good shape and were not changed between the calibration and the field campaign. The blowers. which prevent liquid water to accumulate on the radomes, were switched on all the time. The radar pointing was evaluated by checking the levels at the beginning and the end of the campaign. The levels shown that the radar was pointing almost perfectly vertically. However, since the vertical alignment was not monitored constantly, the spectral and Doppler velocity data should be interpreted carefully, especially in case of strong horizontal winds."

8. P.6, Gas attenuation correction: This description is not entirely clear to me. I understand that you calculate attenuation due to the main attenuating gases at W-Band, which are water vapour and oxygen. I understand that you use the radio sonde profiles to estimate the gas attenuation profile but are you using a constant profile for all the times between two launches or do you interpolate the RS profiles in between? I didn't check the data files but I suggest to include all correction, such as gas attenuation profiles as additional arrays if the raw reflectivity profiles are not provided. It allows the user to retrieve the radar profiles without any of your corrections applied and so the user can maybe also apply his own corrections. The comparison with Dias Neto et al. makes only sense if the columnar water vapour amounts had similar range for your campaign. Did they?

Initially, we only computed the attenuation at the time of the sounding to have an estimate of the order of magnitude, but we did not apply any gas attenuation correction on the data themselves. We now decided to correct for gas attenuation for all time-steps using PAMTRA. For that we computed a linear interpolation between each three-hourly radio-sounding to get the columnar water vapour and other variables needed for the computation of attenuation. To check that this interpolation makes sense, we computed a variogram of the integrated water vapour from the radiosounding (see Fig. 1). We can see that the decorrelation time is long enough so we can expect relatively accurate interpolated values in between radiosoundings. The reflectivity with gas attenuation correction was added to the files as a new variable, while the « raw » reflectivity was left unchanged. This let the user the choice to use our attenuation correction or to apply an other method. We added this information in the manuscript in Sect. 3.3.1.:

"To correct for attenuation due to atmospheric gases, we used the Passive and Active MicrowaveTRansfer Model (PAMTRA Mech et al., 2020) available at https://github.com/igmk/pamtra (last access: May 27th, 2020) and humidity, temperature and pressure profiles from radiosoundings launched at DGW. Radiosoundings were usually available every three hours, but sometimes up to twelve hours. In order to quantify the temporal variability, we computed the

variogram\_of IWV from all radiosoundings and concluded that the decorrelation time is long enough so we can expect relatively accurate interpolated values in between radiosoundings. We hence decided to compute a linear interpolation between the two nearest radiosoundings in time to get the profiles at a 5 min resolution, which was then used to compute the gas attenuation and applied to each WProf profile. "

Figure 1: Variogram of integrated water vapour over the whole ICE-POP 2018 campaign. The red line shows the fit of the spherical variogram model.

**10. P.6, L. 16: Correct citation is "Dias Neto et al., 2019"**

**We corrected it.**

11. Fig. 6 and similar figures for the other cases: The choice of color table is certainly always a matter of taste but I am wondering whether you considered to use a color table for reflectivity and the diameters which contains more than the three colors that you have now. I could image this could show much more structural information. Maybe you can also adjust/reduce the Doppler velocity limits to show better the slight changes from unrimed to rimed snow, which I would guess are often only between 1 and 2.5 m/s. If you extract ZDR from the X-Pol, did you consider to calculate dual-wavelength ratios between X and W-Band? It might be quite noisy due to the strongly different volumes but in stratiform snow it might still provide some useful information.

Thank you for these suggestions. We changed the colormaps for reflectivity to a perceptually uniform colormap with more colors. We also slightly changed the Doppler velocity one and reduced the minimum and maximum values.

We did compute dual-wavelength ratios between X and W bands but with a vertically pointing X-band radar collocated with WProf. This makes it much easier to compare, since the volumes are more similar. We did not want to include that in the manuscript, since the vertically pointing X-band data are not from our radar and we do not have the right to publish it. However, all ICE-POP data should be published at some point. We added a note in the manuscript stating this possibility for future research on p.10 L.32-33: "Future studies could use the data presented in this paper together with other measurements from ICE-POP 2018. This includes radar data at X, Ku and Ka band and is particularly suited for microphysical studies with multi-frequency measurements."

12. Caption Fig 4: Full Stop missing at the end of caption.

We corrected it.

13. P.6, L27: Space missing after comma and before "2 km"

We corrected it.

14. Table 3: The ":" separating hours and minutes is missing on several entries.

We corrected it.

15. Fig. 5: Write out that "dam" means decameters, it might not be a familiar "unit" to all readers.

We corrected it.

16. Fig.6: Again, in the current color scale of the W-band, the variations of mean Doppler velocity, which one could expect due to riming are extremely hard to see. I suggest to experiment more with other color scales in order to better visualize those structures.

As written in the answer to comment 11, we adapted both the colorbar and the limits to make those structures more visible.

17. Fig. 6: It might be worth mentioning in the caption that the 2DVD was measuring at a different location and much lower altitude. This also avoids confusion that the 2DVD was really only measuring rain and not snow (although it could measure snow as well). In this current multi-panel plot it gives the impression all measurements would be from the same location.

We added this information explicitly in the caption and in the text.

18. P 7, L. 23: Is the degree of riming set to zero for the rain cases in the data? Maybe this would be good to avoid confusion for users. At least it should be mentioned in the meta data if not already included.

No, it's usually very high : the raindrops appear as a small bright spot due to the reflection of the flash and this leads to a high degree of riming. Raindrops are classified as small particles. Note that the degree of riming of small particles is also not reliable, since it is computed over a few pixels. We did not take small particles into account to plot the degree of riming, which then also excludes raindrops.

This information has been added in the manuscript: P.4 L.30-32: "Note that the degree of riming of melting particles and raindrops computed by this method is usually quite high. One should not consider the degree of riming for particles with a melting probability higher than 50 %."

19. P. 8, L.1: Maybe better "At 00:00 UTC on 28 February, the nimbostratus."

We corrected it.

20. P. 8, L. 2: How do you know that this is a fog layer and not for example a thin mixed- phase cloud, or drizzle cloud?

Fog is by definition a cloud at or near the surface. It could be both a thin mixedphase cloud or a drizzle cloud. For this reason we think that fog is more general, since it encompasses all clouds at or near the surface and this is the only thing we are sure of looking at the data.

21. Fig. 6 and following radar multi-panel plots: I would find it interesting to have temperature isotherms overplotted in one of the graphs. If you don't have this information, then maybe juts plot the surface temperature. I find it quite difficult to interpret the data without this essential variable.

Thank you for this suggestion. We added the surface temperature on Fig. 6a and similar one for the other events. We preferred showing surface temperature, rather than isotherms interpolated from reanalyses, which may be quite erroneous in such a complex terrain.

22. Fig. 8: I am surprised that the mean Doppler velocities in the W-Band are so similar during periods when the MXPol classifies predominantly aggregates and rimed particles. Shouldn't the vertically pointing Doppler velocities be much faster if they are really rimed? At which rime mass fraction does the MXPol classification detect them as rimed? Isn't it also surprising that the MASC derived rime mass fraction is not really correlating with the MXPol classification? Are you able to provide an error bar for the rime mass fraction?

The period of riming identified by MXPol also correspond to period with intense turbulence and updrafts. In that case, the vertical Doppler velocity from WProf is dominated by vertical wind more than by the microphyiscs. This is why it does not react to riming. It is not possible to determine the rime mass fraction corresponding to the MXPol classification, since it is only base on dual-polarisation data. More information can be found in Besic et al. 2016, 2018. Also be aware that the MXPol classification is only available above 2000 m and hence does not show what is taking place below, as opposed to WProf.

23. Fig. 10: I think you should somehow clearly mark time periods with rain versus others with snow. Especially since the MASC is deriving rime mass fraction for rain drops were it should be set to zero. It would be good if this correction/flagging of the MASC data for rain is done by you in order to avoid somebody is using it in a wrong way.

We added the melting probability in the MASC plots to give an information on the precipitation type.

24. P. 10, L. 4: Change K-u, K-a into common "Ku" and "Ka"

We corrected it.

**Anonymous Referee #2**

Received and published: 27 November 2020

1. The paper titled "Radar and ground-level measurements of precipitation collected by EPFL during the ICE-POP 2018 campaign in South-Korea by Gehring et al" describes the four-month dataset of precipitation and cloud measurements collected during ICE-PO 2018. While I believe the paper is well written, I am not sure if the authors made an attempt to convince a non-expert in this subject as to how important this data set is to the scientific community. I also see that the author is already published an article in ACP partly in regard to this data set focusing on one particular event of 28 February 2018. May be including a little bit more on other available data sets in this category and why these measurements are important compared to other measurements may help the readers who are not experts in this subject. While this dataset seem to be unique. I believe the authors can do a better job in analyzing the available data and may be comparing with other available measurements. Otherwise, this is just another data set. I, therefore recommend authors to do further analyses to show the uniqueness of the data set and it would be helpful for the readers if the authors could explain this data set a bit more with more analyses such as a relationship between precipitation and reflectivity for both W-band and X-band radars.

Thank for your constructive comment on the manuscript. We added information on other available dataset in the introduction to put the data presented in this paper in a broader context. We also strengthened the uniqueness of this dataset and showed some examples of analysis that can be performed. However, we think that analysing the data in more details and in particular comparing with other available measurements is out of the scope of the ESSD journal, which states: "Any interpretation of data is outside the scope of regular articles. [...] Any comparison to other methods is beyond the scope of regular articles." (https://www.earth-system-science-data.net/about/aims\_and\_scope.html)

Here are my comments:

- 2. Line 6, page 1: please use a period following "video disdrometer (https://doi.pangaea.de/10.1594/PANGAEA.918315, Gehring et al. (2020a))" Thank you, we corrected it.
- 3. Line 15, page 2: It may be helpful to add something about how unique your measurements are and the importance of the data to the scientific community particularly to microphysics and their use in evaluating models. Thank you for this comment. We substantially modified the introduction by adding references to other available dataset of similar field campaigns. We also added a note on how this dataset is unique and relevant to the scientific community on P3, L.3: "Such a dataset is unique, as it includes multi-frequency radar and ground-based measurements in a region where similar measurements were scarce before ICE-POP 2018. As shown in Gehring et al. (2020b) it is a useful dataset for snowfall microphysics studies and could also be relevant for validation of numerical weather prediction models".
- 4. Line 23, page 2: Isn't this figure same as figure 1 of Gehring et al., 2020 paper? Almost, we added the location of the 2DVD, since we are using it in this study. We think it is relevant to add this figure instead of citing it, since it is important to have

the location of the instruments in this paper. We added in the caption of Fig. 1 "Adapted from Gehring et al. 2020", to make it clear that we are adapting a figure that is already published.

5. Line 1, page 7: The plots you show for these events (25 November 2017, 28 February 2018, 04 and 07 March 2018) are just a description of what you see in your data. It would be helpful if you could explain this data set a bit more with more analyses such as some sort of relationship between precipitation and reflectivity for both W-band and X-band radars and may be comparing the reflectivity with some other precipitation measurements that are available. I think this would add significance of this data set and the paper.

Thank you for this comment. We added rain rate measurements in MHS in Figs 6, 8, 10, 12 to compare with reflectivity measurements. We shortly discuss the rain rate in the respective sections and its relation to the reflectivity. However, it is out of the scope of the ESSD journal to go into more detailed analyses of the measurements.

6. Line 15, page 7: Since this is the third-largest precipitation accumulation in the record, it may be helpful to see a precipitation rate time series plot and temperature if possible. May be if possible it will be appropriate to do the same sort of analyses for all the main events listed here.

As stated above, we included precipitation rate and temperature at MHS in Figs 6, 8, 10, 12. We discuss the relationship with reflectivity in the respective sections.

 Line 26-32, page 8: I think it may be appropriate to do further analyses here as well as commented above.
 We commented this section a bit more, but again the scope of ESSD is not to go into too dotailed analyses of the data. We refer the reader to Cobring et al. 2020, for

into too detailed analyses of the data. We refer the reader to Gehring et al. 2020, for an in-depth analysis of part of this dataset.

8. Once additional analyses are done, I would revise the "conclusions" section a little bit to include the uniqueness of the measurements and how they compare with other measurements (if they are available). Some sort of validation of the data may help strengthen the paper and the robustness of the data set I believe.

We strengthened the uniqueness of this dataset in the conclusion and included some suggestions of comparisons with other datasets and studies in different mountainous regions on P.11, L.4: "The dataset is unique as it represents one of the first observations of cloud and precipitation with radars at different frequencies and ground-based in situ measurements in the Taebaek mountains. It is complementary to similar dataset in other regions and allows to compare snowfall microphysical studies in different locations. In particular, it is relevant to validate conceptual models of orographic precipitation drawn for other mountain chains such as the studies of Houze and Medina (2005); Panziera et al. (2015); Grazioli et al. (2015)."

A validation of the data presented here is not in the scope of this paper. However the consistency between the measurements from different instruments presented here and in Gehring et al. 2020 gives us a good confidence in this dataset.

Josué Gehring1, Alfonso Ferrone1, Anne-Claire Billault–Roux1, Nikola Besic2, Kwang Deuk Ahn3, GyuWon Lee4, and Alexis Berne1

[revised manuscript text omitted]